# A General Framework for Learning under Corruption: Label Noise, Attribute Noise, and Beyond

## Abstract

Corruption is frequently observed in collected data and has been extensively studied in machine learning under different corruption models. Despite this, there remains a limited understanding of how these models relate such that a unified view of corruptions and their consequences on learning is still lacking. In this work, we formally analyze corruption models at the distribution level through a general, exhaustive framework based on Markov kernels. We highlight the existence of intricate joint and dependent corruptions on both labels and attributes, which are rarely touched by existing research. Further, we show how these corruptions affect standard supervised learning by analyzing the resulting changes in Bayes Risk. Our findings offer qualitative insights into the consequences of "more complex" corruptions on the learning problem, and provide a foundation for future quantitative comparisons. Applications of the framework include corruption-corrected learning, a subcase of which we study in this paper by theoretically analyzing loss correction with respect to different corruption instances.

## 1   Introduction

Machine learning starts with data. The most widespread conception of data defines them as atomic facts, perfectly describing some reality of interest [1]. In learning theories, this is reflected by the often-used assumption that training and test data are drawn independently from the same distribution. The goal of learning is to identify and synthesize patterns based on the knowledge, or information, embedded in these data. In practice, however, corruption regularly occurs in data collection. This creates a mismatch between training and test distributions, forcing us to learn from imperfect facts.

We should thus doubt the view of data as static facts, and consider them as a dynamic element of a learning task [2]. Besides the predictor and the loss function, one may focus on the data dynamics, studying corruptions and intervening in the learning process. Toward this goal, there has been a surge of research in the machine learning community proposing various corruption models, examining and correcting their effects on learning formally or empirically [3, 4, 5, 6, 7, 8]. Nevertheless, it is still unclear how these models relate and whether they characterize all types of corruption. Even though the necessity of investigating this topic is recognized both at a practical [9, 10] and a theoretical [11, 12] level, no standardized way to model and analyze corruption has been so far created [13].

Our primary objective here is to systematically study the problem of learning under corruption, providing a general framework for analysis. Whilst there have been some existing attempts, certain limitations persist in terms of homogeneity and exhaustiveness. A famous early endeavor is Quinonero-Candela et al. [14], grouping together works about the multi-faceted topic of dataset shift, yet not in a unifying or comprehensive manner. Later on, several studies aim to provide a more homogeneous view of corruption, often referred to as noise or distribution shift. However, their frameworks

typically rely on some corruption-invariant assumptions on the marginal or conditional probabilities, and the extent of exhaustiveness is merely conjectured or not considered [15, 16, 17, 18].

In this paper, we take a different point of view from the previous work: we categorize corruption based on its dependence on the feature and label space, rather than relying on the notion of invariance. Our resulting framework is generic, encompassing all possible pairwise stochastic corruptions.[1] The underpinning mathematical tool that enables such exhaustiveness is the Markov kernel. While Markov kernels have been utilized in formalizing corruption [7, 19], their primary focus has been solely on label corruption, attribute corruption, or simple joint corruption. To our knowledge, the proposed framework is novel in the sense of demonstrated exhaustiveness in this domain. Our contributions are summarized as follows:

**C1** We propose a new taxonomy of corruption in the supervised learning setting (§ 3), hierarchically organized through the notion of dependence (Fig. 1), and connect existing corruption models to this taxonomy (Tab. 1).

**C2** We analyze the implications of our family of corruptions on learning (§ 4), linking the Bayes risk of the clean and corrupted supervised learning problems through equality results (Theorem 3, Theorem 4, Theorem 5).

**C3** We derive corruption-corrected loss functions for different corruption instances within our framework (§ 5). A subcase of these corrections (Theorem 8) generalizes prior results on corruption-corrected learning in simple label corruption.

Though abstract in general, our results expand upon existing ones on specific corruption models and shed light on the relatively under-explored joint and dependent corruptions.

## 2 Background

Before introducing our analysis, we review the background framework and notations.

**Supervised learning** In statistical decision theory [20, 21], a general decision problem can be viewed as a two-player game between *nature* and *decision-maker*. Nature chooses its *state*, then *experiment* leads to some *observations* given the state, and the decision-maker picks a suitable *action* from a fixed set of *decision rules*. In the specific setting of *supervised learning*, observations are in the feature space $X \subset \mathbb{R}^d$, $d \geq 1$, states are in the label space $Y$, then the *experiment* $E$ leads to a probability associated with the observation $X$, given the state $Y$. Here we focus on the classification task, that is, assuming the label space to be *finite*. All the stated results can be easily extended to regression cases by considering a continuous label space; we leave it for future application.

To formalize the processes described above, we introduce the Markov kernel.

**Definition 1** (Klenke [22])**.** *A Markov kernel $\kappa$ from a measurable space $(X_1, \mathcal{X}_1)$ to a measurable space $(X_2, \mathcal{X}_2)$ is a function $x_1 \mapsto \kappa(x_1, \cdot)$ from $X_1$ to $\mathcal{P}(X_2)$, the set of probability measures on $X_2$, such that $\kappa(x_1, B)$ is measurable in $x_1$ for each set $B \in \mathcal{X}_2$. We denote it by $\kappa : X_1 \rightsquigarrow X_2$, or more compactly by $\kappa_{X_1 X_2}$. The set of Markov kernels from $X_1$ to $X_2$ is referred to as $\mathcal{M}(X_1, X_2)$.*

The Markov kernel generalizes the concept of conditional probability. Looking at the function $\kappa(\cdot, B)$, it associates different probabilities to the set $B$ given different values of the *parameter* $x_1$. It can transform a distribution $\mu \in \mathcal{P}(X_1)$ into another distribution $\mu\kappa \in \mathcal{P}(X_2)$, as well as transform a function $f : X_2 \to \mathbb{R}$ into another function $\kappa f : X_1 \to \mathbb{R}$ with the following two operators:

$$\mu\kappa(B) := \int_{X_1} \kappa(x_1, B)\mu(dx_1) \ \forall B \in \mathcal{X}_2 , \qquad \kappa f(x_1) := \int_{X_2} \kappa(x_1, dx_2)f(x_2) \ \forall x_1 \in X_1 ,$$

provided the integral exists. Next, we define different operations to combine Markov kernels:

**P1** Given $\kappa : X_1 \rightsquigarrow X_2$ and $\lambda : X_1 \times X_2 \rightsquigarrow X_3$, their *chain composition* $\kappa \circ \lambda : X_1 \rightsquigarrow X_3$ is defined by $(\kappa \circ \lambda)f(x_1) := \int_{X_2} \kappa(x_1, dx_2) \int_{X_3} \lambda((x_1, x_2), dx_3)f(x_3) = \kappa(\lambda\, f)(x_3)$ where $f : X_3 \to \mathbb{R}$ is a positive $\mathcal{X}_3$-measurable function;

**P2** For $\kappa : X_1 \rightsquigarrow X_2$ and $\lambda : X_1 \times X_2 \rightsquigarrow X_3$, their *product composition* $\kappa \times \lambda : X_1 \rightsquigarrow X_2 \times X_3$ is $(\kappa \times \lambda)f(x_1) := \int_{X_2} \kappa(x_1, dx_2) \int_{X_3} \lambda((x_1, x_2), dx_3)\, f(x_2, x_3)$ for every $f$ positive $\mathcal{X}_2 \times \mathcal{X}_3$-measurable.

---

[1] As for non-stochastic ones, we show that they always have a stochastic alternative representation. See § 3.

Notice that a probability distribution is a specific instance of a Markov kernel, constant in its parameter space. Therefore, **P1** and **P2** are well defined for $\kappa \equiv \mu \in \mathcal{P}(X_2)$. We can unify the notation of $\times$ for distributions thanks to the flexibility of kernels, and consider the $\mu\kappa$ as a subcase of $\mu \circ \kappa$.

**Bayes risk**    Having defined all these objects, a supervised learning problem can be represented by the diagram $\ Y \overset{E}{\rightsquigarrow} X \overset{h}{\rightsquigarrow} Y, \ $ where $h$ is a decision rule chosen in $\mathcal{M}(X, Y)$. Its task can be formalized as a risk minimization problem, i.e., finding the optimal action $h \in \mathcal{H}$ by considering the *Bayes Risk* (BR) measure

$$BR_\ell(\pi \times E) = \inf_{h \in \mathcal{M}(X,Y)} R_{\pi,\ell}(\pi \times E) = \inf_{h \in \mathcal{M}(X,Y)} \mathbb{E}_{\mathsf{Y} \sim \pi} \mathbb{E}_{\mathsf{X} \sim E_\mathsf{Y}} \ell(h_\mathsf{X}, \mathsf{Y}) \ ,$$

where the notation $\kappa_\mathsf{X}$ stands for the kernel $\kappa$ evaluated on the parameter X, e.g., $h_\mathsf{X}, E_\mathsf{Y}$ (this subscript notation will be used throughout), and $\pi$ is a prior distribution on $Y$. The function $\ell$ is asked to be bounded and a *proper loss* [23, 24], i.e., a loss function $\ell : \mathcal{P}(Y) \times Y \to \mathbb{R}^+$ whose minimization set contains the ground truth class probability. More formally, we ask for

$$\exists \, h^* \in \arg \min_{h \in \mathcal{M}(X,Y)} R_{\pi,\ell,\mathcal{A}}(E) \text{ such that } h^* \times \mu = E \times \pi \, , \exists \, \mu \in \mathcal{P}(X) \ .$$

Since in real-world applications, one deploys a model with only limited representation capacity, we consider the constrained version of BR

$$BR_{\ell,\mathcal{H}}(\pi_Y \times E) = \inf_{h \in \mathcal{H} \subseteq \mathcal{M}(X,Y)} \mathbb{E}_{\mathsf{Y} \sim \pi_Y} \mathbb{E}_{\mathsf{X} \sim E_\mathsf{Y}} \ell(h_\mathsf{X}, \mathsf{Y}) \ .$$

We call $\mathcal{H}$ the *model class*. If we fix the joint space to $Z = X \times Y$ and the joint probability distribution to $P = \pi_Y \times E \in \mathcal{P}(Z)$, we can refer to a *supervised learning problem* as the triple $(\ell, \mathcal{H}, P)$. Notice that we can also use an equivalent decomposition of the joint distribution through a posterior kernel $F : X \rightsquigarrow Y$, so that $P = \pi_X \times F$ for some prior on the feature space. Hence, each supervised learning problem can have two associated kernels, the experiment $E$ and the posterior one $F$. We then obtain two views of the learning problem, a generative and a discriminative one, as previously noted by Reid et al. [25]. By means of these, we can define two *Conditional BR* (CBR):

$$\text{Discriminative: } \mathbb{E}_{\mathsf{X} \sim \pi_X} CBR_{\ell,\mathcal{H}}(F_\mathsf{X}) = \mathbb{E}_{\mathsf{X} \sim \pi_X} \inf_{h_\mathsf{X} \in \mathcal{H}_\mathsf{X}} \mathbb{E}_{\mathsf{Y} \sim F_\mathsf{X}} \ell(h_\mathsf{X}, \mathsf{Y}) \ , \tag{1}$$

$$\text{Generative: } \mathbb{E}_{\mathsf{Y} \sim \pi_Y} CBR_{\ell,\mathcal{H}}(E_\mathsf{Y}) = \mathbb{E}_{\mathsf{Y} \sim \pi_Y} \inf_{h \in \mathcal{H}} \mathbb{E}_{\mathsf{X} \sim E_\mathsf{Y}} \ell(h_\mathsf{X}, \mathsf{Y}) \ ,$$

both equal to their corresponding constrained BR. Notice that for Eq. (1) to be well defined, we need at least one minimum of the unconstrained BR to be included in the model class. For our convenience, we ask it to be the $h$ *matching* the $F$.

# 3  A general framework for corruption

In this section, we present a general framework of pairwise corruptions based on the notion of *dependence* and discuss how existing corruption models fit into this framework as subcategories. First, let us formally define corruption and two additional kernel operations, which will be useful in the buildup of our corruption taxonomy.

**Definition 2.** *A corruption is a Markov kernel $\kappa$ that sends a probability space $(X \times Y, \mathcal{X} \times \mathcal{Y}, P)$ into another, $(X \times Y, \mathcal{X} \times \mathcal{Y}, \tilde{P})$. We write it as $\kappa_{Z\tilde{Z}}$,[2] and call the variables $z = (x, y) \in Z$ parameters and the differentials $d\tilde{z} = d\tilde{x}d\tilde{y}$ corrupted variables.*

The following operations are not considered in the classical probability literature but have been studied in other areas, e.g., through the lens of category theory [26, 27, 28]. Here we rework them to fit our framework.

**P3** Given $\kappa : X_1 \rightsquigarrow X_2$ and $\lambda : X_3 \rightsquigarrow X_4$, their *superposition* (see § S2.1) is equal to $\kappa\lambda : X_1 \times X_3 \rightsquigarrow X_2 \times X_4$ as $(\kappa\lambda)f(x_1, x_3) \coloneqq \int_{X_2} \kappa(x_1, dx_2) \int_{X_4} \lambda(x_3, dx_4) \, f(x_2, x_4)$, where $f : X_2 \times X_4 \to \mathbb{R}$ is positive $\mathcal{X}_2 \times \mathcal{X}_4$-measurable;

---

[2]We slightly abuse the kernel notation $\kappa_{Z\tilde{Z}}$ to describe how corruption changes the probability spaces. For instance, if a corruption acts solely on the space $X$, it will be written as $\kappa_{X\tilde{X}}$; however, only the probability measure on it will be actually changed.

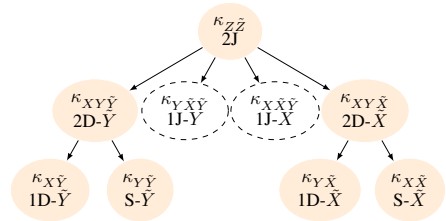 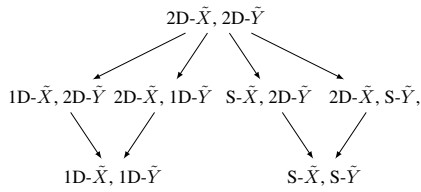

(a) Corruption hierarchy. It is based on the independence from a parameter or a corrupted variable. Arrow: child is constant w.r.t. exactly one of the variables in parent.

(b) Feasible combinations. The partial ordering is induced by corruptions, i.e., one corruption in child and one in parent respect the corruption hierarchy.

Figure 1: Partial orderings on the corruption and combination sets, based on the amount of *dependence* on the spaces. In the left panel, we underline with dotted nodes the corruptions that cannot be used in any feasible combination. Trivial cases of independence from all parameters or identical kernels are excluded from this analysis.

**P4** The *pseudo-inverse* of a kernel $\kappa : X_1 \rightsquigarrow X_2$ is defined as $\kappa^\dagger : X_2 \rightsquigarrow X_1$ such that $(\kappa^\dagger \circ \kappa)\mu_1 = \mu_1$ and $(\kappa \circ \kappa^\dagger)\mu_2 = \mu_2$ with $\mu_1, \mu_2$ being the probabilities associated to $X_1, X_2$. In general, the pseudo-inverse is not unique, since it corresponds to a class of equivalence induced by the probability measure on $X_1$ (see details in § S2.2).

Again, **P3** is well defined for $\kappa \equiv \mu \in \mathcal{P}(X_2)$. This operation allows for more flexible combinations of kernels, in a "parallel" fashion. No restriction is imposed on the parameter spaces to be equal, e.g., $X_1 = X_2$, or Cartesian products with some space in common, e.g., $X_1 = Y_1 \times Y_2, X_2 = Y_1 \times Y_3$. When this happens, the action of the two kernels "superpose" on the same space. In addition, having more than one measure in the integral acting on the same space would make the integral ill-defined, so this case is excluded. Because of these properties, we say that **P3** is the operation with the *weakest feasibility conditions*, i.e., the set of rules to fulfill a well-defined operation.

**Building a taxonomy of corruptions** Corruptions can be naturally classified in different ways, depending on their behavior with respect parameters and corrupted variables. In Fig. 1a, we show all possible non-trivial corruption types, i.e., those that are not identical and not constantly equal to a probability. We classify them based on the number of parameters they *depend on*, and the type of corrupted variables they *result in*. Specifically, we employ the following abbreviations: J is short for Joint (both variables are corrupted), S is short for Simple (the parameter and the corrupted variable are the same), and D is short for Dependent (others). We then obtain the classification: 2-parameter joint corruption (2J), 1-parameter joint corruption (1J), 2-parameter dependent corruption (2D), 1-parameter dependent corruption (1D), simple corruption (S), along with an indication of parameter or corrupted space. The general naming rule is {#parameters} + {abbreviation} + {-} + {parameter or corrupted space, depending on where the ambiguity lies}.

We now want to generate all possible corruptions of the type $\kappa_{Z\tilde{Z}} : X \times Y \rightsquigarrow \tilde{X} \times \tilde{Y}$. We combine the nodes in Fig. 1a using the superposition operation (**P3**), obtaining all the feasible combinations included in Fig. 1b. The missing couples are excluded because of **P3**'s feasibility conditions described above, which, even if weak, still do not allow some corruption pairings. Needing each corrupted variable to appear *exactly once*, we cannot include the 1-parameter joint corruptions in any factorization of the $\kappa_{XY\tilde{X}\tilde{Y}}$. It is easy to check that no corruption from (a subset of) $\{X, Y\}$ to (a subset of) $\{\tilde{X}, \tilde{Y}\}$ can be combined with them. Compatibility problems arise also when trying to combine a simple corruption (S) with a 1-parameter dependent one (1D); we cannot fulfill the feasibility conditions for **P3** and obtain a complete joint corruption, since we will be always missing a parameter. We then exclude this combination from our taxonomy.[3]

**Markov kernels and exhaustiveness** Our motivation for formalizing corruptions through Markov kernels is their representation power in terms of couplings. A *coupling* is formally defined for two probability spaces $\Sigma_1 := (Z_1, \mathcal{Z}_1, P_1)$, $\Sigma_2 := (Z_2, \mathcal{Z}_2, P_2)$ as a probability space $\Sigma := (Z_1 \times Z_2, \mathcal{Z}_1 \times \mathcal{Z}_2, P)$, such that the marginal probabilities associated to $P$ w.r.t. $Z_i$, $i \in \{1, 2\}$ are

---

[3]Note that 1Js are still valid corruptions if seen as a subcase of a 2J, the full one, e.g., $2J = \kappa_{X\tilde{X}\tilde{Y}}(d\tilde{x}d\tilde{y}, x)\mathbf{1}(y)$. Similarly, a 1D corruption can be seen as a subcase of a 2D corruption. Here we are exploring the possibility of combining them with other corruptions. The constraints are only dimensional.

Table 1: Illustration of the taxonomy with examples of existing corruption models.

| Name | Action diagram | Corrupted distribution | Examples |
|---|---|---|---|
| S-$\tilde{X}$ | $Y \overset{E}{\rightsquigarrow} X \overset{\kappa_{X\tilde{X}}}{\rightsquigarrow} \tilde{X}$ | $\tilde{P} = (\kappa_{X\tilde{X}}\delta_{Y\tilde{Y}}) \circ (\pi_Y \times E)$ | attribute noise [30, 31, 4, 19] |
| S-$\tilde{Y}$ | $X \overset{F}{\rightsquigarrow} Y \overset{\kappa_{Y\tilde{Y}}}{\rightsquigarrow} \tilde{Y}$ | $\tilde{P} = (\delta_{X\tilde{X}}\kappa_{Y\tilde{Y}}) \circ (\pi_X \times F)$ | class-conditional noise [32, 33, 5, 34, 7, 19] |
| 1D-$\tilde{X}$ | $X \overset{F}{\rightsquigarrow} Y \overset{\kappa_{Y\tilde{X}}}{\rightsquigarrow} \tilde{X}$ | $\tilde{P} = (\kappa_{Y\tilde{X}}\delta_{Y\tilde{Y}}) \circ (\pi_X \times F)$ | style transfer [35, 36, 37] |
| 1D-$\tilde{Y}$ | $Y \overset{E}{\rightsquigarrow} X \overset{\kappa_{X\tilde{Y}}}{\rightsquigarrow} \tilde{Y}$ | $\tilde{P} = (\delta_{X\tilde{X}}\kappa_{X\tilde{Y}}) \circ (\pi_Y \times E)$ | instance-dependent noise (IDN) [8] |
| 2D-$\tilde{X}$ | $Y \overset{E}{\rightsquigarrow} X \overset{\kappa_{XY\tilde{X}}}{\underset{\kappa_{XY\tilde{X}}}{\rightsquigarrow}} \tilde{X}$ | $\tilde{P} = (\kappa_{XY\tilde{X}}\delta_{Y\tilde{Y}}) \circ (\pi_Y \times E)$ | adversarial noise [38, 39, 40, 41, 42] |
| 2D-$\tilde{Y}$ | $X \overset{F}{\rightsquigarrow} Y \overset{\kappa_{XY\tilde{Y}}}{\underset{\kappa_{XY\tilde{Y}}}{\rightsquigarrow}} \tilde{Y}$ | $\tilde{P} = (\delta_{X\tilde{X}}\kappa_{XY\tilde{Y}}) \circ (\pi_X \times F)$ | instance & label-dependent noise [8, 43, 44, 45] |
| S-$\tilde{X}$, S-$\tilde{Y}$ | $\tilde{Y} \overset{\kappa_{Y\tilde{Y}}}{\rightsquigarrow} Y \overset{E}{\rightsquigarrow} X \overset{\kappa_{X\tilde{X}}}{\rightsquigarrow} \tilde{X}$ | $\tilde{P} = (\kappa_{X\tilde{X}}\kappa_{Y\tilde{Y}}) \circ (\pi_Y \times E)$ | combined simple noise [19] |
| 1D-$\tilde{X}$, 2D-$\tilde{Y}$ | $\tilde{X} \overset{\kappa_{Y\tilde{X}}}{\rightsquigarrow} Y \overset{E}{\rightsquigarrow} X \overset{\kappa_{XY\tilde{Y}}}{\underset{\kappa_{XY\tilde{Y}}}{\rightsquigarrow}} \tilde{Y}$ | $\tilde{P} = (\kappa_{Y\tilde{X}}\kappa_{XY\tilde{Y}}) \circ (\pi_Y \times E)$ | target shift [46, 47, 48, 49] |
| 2D-$\tilde{X}$, S-$\tilde{Y}$ | $\tilde{Y} \overset{\kappa_{Y\tilde{Y}}}{\rightsquigarrow} Y \overset{E}{\rightsquigarrow} X \overset{\kappa_{XY\tilde{X}}}{\underset{\kappa_{XY\tilde{X}}}{\rightsquigarrow}} \tilde{X}$ | $\tilde{P} = (\kappa_{XY\tilde{X}}\kappa_{Y\tilde{Y}}) \circ (\pi_Y \times E)$ | mutually contaminated distributions [6, 50, 51] |
| 2D-$\tilde{X}$, 1D-$\tilde{Y}$ | $\tilde{Y} \overset{\kappa_{X\tilde{Y}}}{\rightsquigarrow} X \overset{F}{\rightsquigarrow} Y \overset{\kappa_{XY\tilde{X}}}{\underset{\kappa_{XY\tilde{X}}}{\rightsquigarrow}} \tilde{X}$ | $\tilde{P} = (\kappa_{XY\tilde{X}}\kappa_{X\tilde{Y}}) \circ (\pi_X \times F)$ | covariate shift [3, 52, 53, 54] |
| 2D-$\tilde{X}$, 2D-$\tilde{Y}$ | $\tilde{Y} \overset{\kappa_{XY\tilde{Y}}}{\underset{\kappa_{XY\tilde{Y}}}{\rightsquigarrow}} Y \overset{E}{\rightsquigarrow} X \overset{\kappa_{XY\tilde{X}}}{\underset{\kappa_{XY\tilde{X}}}{\rightsquigarrow}} \tilde{X}$ | $\tilde{P} = (\kappa_{XY\tilde{X}}\kappa_{XY\tilde{Y}}) \circ (\pi_Y \times E)$ | generalized target shift [55, 56, 57] concept drift [58, 59] |

the respective $P_i$. By construction, Markov kernels are in bijection with all the possible couplings existent on $Z \times Z$ with two *fixed* probability measures, for us, $P, \tilde{P}$. Hence, they represent all possible pairwise dependencies between probability spaces that are *stochastic*, and for non-stochastic mappings, we are sure to have an alternative Markov kernel representation.[4]

In most machine learning research considering corruption, the corruption process typically involves two environments, that is, the training one and the test one. Our definition of corruption (Def. 2) covers all such pairwise cases. Furthermore, one may also apply this framework to settings with more than two spaces, e.g., online learning or learning from multiple different domains [29]. For these cases, we can employ a composed model, where different corruptions are acting together in a "chained" (**P1**, **P2**) or "parallel" (**P3**) fashion and creating more complex patterns. We discuss further possibilities for applying this framework to $n > 2$ corrupted spaces in § S2.3.

**Relations to existing paradigms**  Next, we examine how existing corruption models fit into our taxonomy. To do so, we reformulate them as specific instantiations of Markov corruptions. This reveals their relationships within the corruption hierarchy presented in Fig. 1a. Our goal here is not to merely demonstrate that a child problem can be solved by a parent one, but rather to gain a deeper understanding of the problem settings. The exhaustiveness of the framework allows us to identify what has been previously overlooked in characterizing all types of corruption. Notably, we highlight the existence of joint and dependent corruptions, which receive far less attention than simple

---

[4]When the mapping between two fixed probability spaces is a transition kernel, e.g., a non-normalized Markov kernel, the map is *deterministic*. An example is the selection bias classically formalized as absolutely continuous probabilities $\tilde{P} \ll P$ [14]. However, given the bijection with the coupling space, we can always find a stochastic map connecting $\Sigma_1, \Sigma_2$. A similar argument can be replicated for mappings between $\Sigma_1, \Sigma_2$ that are not kernel-induced, e.g., they are not positive. For more details, see § S2.2.

corruptions, while far greater problems arise in such complicated cases (see § 4). Moreover, we notice that existing categorizations rely mostly on the notion of invariance, i.e., corruptions are defined based on which element of the distributions are preserved. These invariance-based taxonomies have been introduced mainly for robustness and causal analyses. However, they do not have a one-to-one correspondence with ours, and do not allow for a hierarchical nor compositional view of corruption. A summary of representative corruption models in the literature is given in Tab. 1, while all the technical details about correspondences and relations between taxonomies are given in § S1.

# 4 Consequences of corruption in supervised learning

Traditionally, experiments have been compared through Bayes Risk using what is known as the Data Processing Inequality, or Blackwell-Sherman-Stein Theorem [20, 60].[5] Recently, in Williamson and Cranko [19], Data Processing Equality results have also been studied within the supervised learning framework. Here we adopt the equality approach to compare the clean and corrupted experiments through Bayes Risk. The equalities formally characterize how the optimization problem is affected by the different kinds of joint corruption in our taxonomy. This gives us a quantitative result in terms of conserved "information" [19] between corrupted and clean learning problems, and a bridge between the problems themselves.

We rewrite the minimization set of the BR in a more compact way, such as $\ell \circ \mathcal{H} := \{(x, y) \mapsto \ell(h_x, y) \mid h \in \mathcal{H} \subseteq \mathcal{M}(X, Y)\}$. We define the action of a corruption $\kappa$ on this set as the set of all the corrupted functions $\kappa f$, $f \in \ell \circ \mathcal{H}$. Lastly, we ask $f^* = \ell \circ h^* \in \arg\min_f \mathbb{E}_{\tilde{P}}[f(\tilde{X}, \tilde{Y})]$ to belong to the constraining space $\ell \circ \mathcal{H}$, for reasons already discussed for Eq. (1).

The first two theorems cover the (S, 2D) cases and their subcase (S-$\tilde{X}$, S-$\tilde{Y}$), as proved in [19].

**Theorem 3** (BR under (S-$\tilde{X}$, S-$\tilde{Y}$), (2D-$\tilde{X}$, S-$\tilde{Y}$) joint corruption). *Let* $(\ell, \mathcal{H}, P)$ *be a learning problem,* $E : Y \rightsquigarrow X$ *an experiment and* $\kappa^{\tilde{X}} \in \{\kappa_{X\tilde{X}}, \kappa_{YX\tilde{X}}\}$ *a corruption. Let* $\kappa_{Y\tilde{Y}}$ *be a simple corruption on* $Y$. *Then we can form the corrupted experiment as per the transition diagram*[6]

$$\tilde{Y} \overset{\kappa_{Y\tilde{Y}}}{\rightsquigarrow} Y \underset{\underset{\kappa^{\tilde{X}}}{\smile}}{\overset{E}{\rightsquigarrow}} X \overset{\kappa^{\tilde{X}}}{\rightsquigarrow} \tilde{X} \quad \text{and obtain}$$

$$\mathbb{E}_{\tilde{Y} \sim \kappa_{Y\tilde{Y}} \pi_Y} CBR_{\ell \circ \mathcal{H}}\big(\kappa^{\tilde{X}} E_{\tilde{Y}}\big) = \mathbb{E}_{Y \sim \pi_Y} CBR_{\kappa^{\tilde{X}}(\kappa_{Y\tilde{Y}} \ell \circ \mathcal{H})}\big(E_Y\big) \,.$$

*Moreover, if* $\kappa^{\tilde{X}} = \kappa_{X\tilde{X}}$, *we have*

$$BR_{\ell \circ \mathcal{H}}[\kappa_{Y\tilde{Y}}(\pi_Y \times \kappa_{X\tilde{X}} E)] = BR_{\kappa_{X\tilde{X}}(\kappa_{Y\tilde{Y}} \ell \circ \mathcal{H})}(\pi_Y \times E) \,. \tag{2}$$

Here in Theorem 3 we have shown the BR equality for the experiment $E$, in line with the Comparison of Experiments and Information Equalities literature mentioned at the beginning of the section. However, for some corruptions the equalities results cannot be stated with $E$ and the Generative CBR, unless ignoring the joint corruption factorization formula (see § S5 for a detailed explanation). We hence use the posterior kernel $F$ defined with the Discriminative CBR (Eq. (1)), and gain more insights about the minimization set while paying a price in elegance of the result.

**Theorem 4** (BR under (S-$\tilde{X}$, 2D-$\tilde{Y}$) joint corruption). *Let* $(\ell, \mathcal{H}, P)$ *be a learning problem,* $F : X \rightsquigarrow Y$ *a posterior and* $\kappa_{XY\tilde{Y}}\}$ *a* $Y$ *corruption. Let* $\kappa_{X\tilde{X}}$ *be a simple corruption on* $X$. *Then we can form the corrupted experiment as per the transition diagram* $\quad \tilde{X} \overset{\kappa_{X\tilde{X}}}{\rightsquigarrow} X \underset{\underset{\kappa^{\tilde{Y}}}{\smile}}{\overset{F}{\rightsquigarrow}} Y \overset{\kappa_{XY\tilde{Y}}}{\rightsquigarrow} \tilde{Y}$

*and obtain*

$$\mathbb{E}_{\tilde{X} \sim \kappa_{X\tilde{X}} \pi_X} CBR_{\ell \circ \mathcal{H}}(\kappa_{XY\tilde{Y}} F_{\tilde{X}}) = \mathbb{E}_{X \sim \pi_X} CBR_{\kappa_{X\tilde{X}}(\kappa_{XY\tilde{Y}} \ell \circ \mathcal{H})}(F_X) \,.$$

---

[5]Briefly, the theorem states that for an experiment $E$ and its image through a suitably defined Markov kernel $\kappa$ w.r.t. some operation, we have $BR_{\pi, \ell, \mathcal{H}}(E) \leq BR_{\pi, \ell, \mathcal{H}}(\kappa E)$ for all $\pi, \ell, \mathcal{H}$.

[6]The first arrow in the diagram is $\tilde{Y} \rightsquigarrow Y$, the opposite direction given for the $Y$ corruption. However, we are not using any notion of inverse corruption here. We are only using the flexibility of Markov kernels as operators and introducing an alternative notation. The kernel used here is exactly the $\kappa_{Y\tilde{Y}} : Y \rightsquigarrow \tilde{Y}$, which acts on an input measure in a "push-forward" fashion. The notation will be further used in the rest of the paper.

We can notice, thanks to Theorems 3, 4, that when corruption involves dependent structures in the factorization, the loss function or the whole minimization set are modified in a parameterized, *dependent* way. For instance,

$$\kappa_{X\tilde{X}}(\kappa_{XY\tilde{Y}}\ell \circ \mathcal{H}) = \{\kappa_{X\tilde{X}}(\kappa_{Y\tilde{Y}}\ell_x \circ h) \mid h \in \mathcal{H}\} \,,$$

with $\kappa_{XY\tilde{Y}}$ now viewed as a parameterized label corruption, i.e. $(\kappa_{Y\tilde{Y}})_x$. An additional consequence is also that the result can only be given in terms of CBR, Discriminative or Generative. We also see that corruptions on $Y$ only affect the loss function and does not touch the model class, even in the dependent case.

The next theorems cover the factorizations involving 1D corruptions. In the first case, we are again forced to use either $E$ or $F$, depending on the involved factors. We group the two results in one theorem for brevity.

**Theorem 5** (BR under (1D, 2D) joint corruption). *Let $(\ell, \mathcal{H}, P)$ be a learning problem, $E : Y \rightsquigarrow X$ and $F : X \rightsquigarrow Y$ be an experiment and a posterior on it.*

*1. Let $\kappa_{Y\tilde{X}}$ be a corruption on $X$ and $\kappa_{XY\tilde{Y}}$ be a corruption on $Y$, then we can form the jointly corrupted experiment as per the transition diagram $\tilde{X} \xrightarrow{\kappa_{Y\tilde{X}}} Y \xrightarrow{E} X \xrightarrow{\kappa_{XY\tilde{Y}}} \tilde{Y}$ and obtain*

$$BR_{\ell \circ \mathcal{H}}[\kappa_{Y\tilde{X}}\kappa_{XY\tilde{Y}}(\pi_Y \times E)] = \mathbb{E}_{\mathsf{Y} \sim \pi_Y} CBR_{\kappa_{Y\tilde{X}}(\kappa_{XY\tilde{Y}}\ell \circ \mathcal{H})}(E_{\mathsf{Y}}) \,. \tag{3}$$

*2. Let $\kappa_{X\tilde{Y}}$ be a corruption on $Y$ and $\kappa_{XY\tilde{X}}$ be a corruption on $X$, then we can form the jointly corrupted posterior as per the transition diagram $\tilde{Y} \xrightarrow{\kappa_{X\tilde{Y}}} X \xrightarrow{F} Y \xrightarrow{\kappa_{XY\tilde{X}}} \tilde{X}$ and obtain*

$$BR_{\ell \circ \mathcal{H}}[\kappa_{X\tilde{Y}}\kappa_{XY\tilde{X}}(\pi_X \times F)] = \mathbb{E}_{\mathsf{X} \sim \pi_X} CBR_{\kappa_{XY\tilde{X}}(\kappa_{X\tilde{Y}}\ell \circ \mathcal{H})}(F_{\mathsf{X}}) \,. \tag{4}$$

Being the (1D, 1D) a subcase of both previous corruptions, we can prove the result as a simple corollary. Notice that this implies both $E$ and $F$ formulations to hold.

**Corollary 6** (BR under (1D, 1D) joint corruption). *Let $(\ell, \mathcal{H}, P)$ be a learning problem, $E : Y \rightsquigarrow X$ and $F : X \rightsquigarrow Y$ be an experiment and a posterior on it. Let $\kappa_{Y\tilde{X}}$ be a corruption on $X$ and $\kappa_{X\tilde{Y}}$ be a corruption on $Y$, then we can form the jointly corrupted experiment as per the transition diagram $\tilde{X} \xrightarrow{\kappa_{Y\tilde{X}}} Y \xrightarrow{E} X \xrightarrow{\kappa_{X\tilde{Y}}} \tilde{Y}$ or equivalently $\tilde{Y} \xrightarrow{\kappa_{X\tilde{Y}}} X \xrightarrow{F} Y \xrightarrow{\kappa_{Y\tilde{X}}} \tilde{X}$. We obtain*

$$BR_{\ell \circ \mathcal{H}}[\kappa_{Y\tilde{X}}(\pi_Y \times \kappa_{X\tilde{Y}}E)] = BR_{\kappa_{Y\tilde{X}}(\kappa_{X\tilde{Y}}\ell \circ \mathcal{H})}(\pi_Y \times E) \,,$$

*or equivalently*

$$BR_{\ell \circ \mathcal{H}}[\kappa_{X\tilde{Y}}(\pi_X \times \kappa_{Y\tilde{X}}F)] = BR_{\kappa_{Y\tilde{X}}(\kappa_{X\tilde{Y}}\ell \circ \mathcal{H})}(\pi_X \times F) \,.$$

In all the Theorems involving a 1D corruption, the minimization set is heavily modified. In Eq. (3), the loss function is corrected such that it will be dependent on the parameter $x$ ($\ell_x$), while the whole composition will be evaluated on $y$ instead of $x$. We the obtain functions of the form $\widetilde{\ell_x \circ h}(y)$. In Eq. (4), we instead end up having a minimization space of the form $\widetilde{(\ell \circ h)}_y(x)$. Lastly, both results of Corollary 6 lead to a comparison of performance on the $X$ space instead of $Y$, with a new loss function that takes in imput $y$ and a probability on $X$ parameterized by $y$. We can consider the these cases as an expansion of the loss space; more detail will be added in the next section.

The only factorization missing from Fig. 1b is the (2D, 2D) one. Because of its high dependence on the parameters, we could not recover a meaningful decomposition of the effect on $\ell \circ \mathcal{H}$. This suggests it to be equivalent to a 2J corruption when looked at through the lens of Bayes Risk. For detailed analysis, see Supplementary material § S3.

## 5 Corruption-corrected learning

We now leverage our corruption framework for answering the question "*what can we do to ensure unbiased learning from biased data?*". This question has different answers depending on what we mean by unbiased learning. As for the biased data, we assume that biased here refers to non-identical joint corruption acting on a probability, giving us a corrupted training distribution.

Past work from Van Rooyen and Williamson [7] and Patrini et al. [34] considered unbiased learning as what is known as *generalization*, i.e. learn on the corrupted space $\tilde{P}$ a hypothesis $h^*$ such that it is also optimal on the clean distribution $P$ at test time. They choose the approach of corrected learning, which is, correcting the loss function or the model class in order to learn a $h^*$ capable to generalize. They both used frameworks related to ours, although only in the presence of simple $Y$ corruption.

We prove similar results to these works for the *loss correction* task and analyze what we can achieve in other corruption cases described by our taxonomy. In general, we cannot prove generalization but we exhibit a corrected loss allowing the model learned on $\tilde{P}$ to have the same *biases (i.e. loss scores)* as the one found for the clean learning problem. To do so, we make use of the pseudo-inverse of a Markov kernel (**P4**), as it is more convenient and powerful than the kernel reconstruction introduced in [7]. The results we show here also serve as a first step towards understanding the effect of corruption of the minimization set $\ell \circ \mathcal{H}$, in the cases where the BR equalities are not giving us much information (i.e. all the cases that are not simple label noise [19]).

Again, in this analysis, we ignore the influence of the data sample and the optimization technique. We use all the assumptions introduced when defining the learning problem in § 2 and the BR results § 4.

**The BR equalities for cleaning kernels**     The theorems proved in § 4 can then be restated, in terms of learning problems and pseudo-inverse $\kappa^\dagger : \tilde{Z} \rightsquigarrow Z$, as

$$(\ell, \mathcal{H}, P) \to (\kappa^\dagger(\ell \circ \mathcal{H}), \kappa P) . \tag{5}$$

We will refer here to the pseudo-inverse of our corruption as the *cleaning kernel*. Notice that the set $\kappa^\dagger(\ell \circ \mathcal{H})$ is not trivially decomposable as $\tilde{\ell} \circ \tilde{\mathcal{H}}$ for some loss and model class. In this case, $\kappa^\dagger(\ell \circ \mathcal{H})$ is said to have no ∘-*factorized structure*.

The BR equalities are ensuring the existence of a function $f^* \in \ell \circ \mathcal{H} \ \cap \ \kappa^\dagger(\ell \circ \mathcal{H})$ that minimizes the Bayes Risk, i.e.

$$f^* \in \underset{f \in \ell \circ \mathcal{H}}{\arg\min} \, \mathbb{E}_P f(Z) \text{ and } f^* \in \underset{f \in \kappa^\dagger(\ell \circ \mathcal{H})}{\arg\min} \, \mathbb{E}_{\kappa P} f(Z) .$$

Sadly, this is not enough for us to find an optimal hypothesis working for both probability spaces. Formal results on this optimal $h \in \mathcal{H}$ for both clean and corrupted spaces only exist for label noise [7, 8]. However, by introducing a few further assumptions, we can get results on which alternative loss to use on train distribution so that the learned $h$ on $\tilde{P}$ will have the same performance scores as the optimal on $(\ell, \mathcal{H}, P)$. Let us consider the composed representation of the function $f^*$ in the test (clean) minimization set, which is $f^* = \ell \circ h^*$. We want to construct a suitable composed representation for $f^*$ also in the space $\kappa^\dagger(\ell \circ \mathcal{H})$, namely $f^* = \tilde{\ell} \circ \tilde{h}^*$. We start by fixing a $\tilde{h}^* \in \mathcal{H}$ of our choice, that if asked to be invertible (**A1**) identifies the loss function as $\tilde{\ell} = \ell \circ h^* \circ (\tilde{h}^*)^{-1} : \mathcal{P}(Y) \times Y \to \mathbb{R}^+$.[7] There can be weaker conditions on $(\tilde{\ell}, \tilde{h}^*)$ enabling all the following results, but do not investigate the here.

Since in general $\kappa^\dagger(f^*) \neq f^*$, we have that: $\exists h' \in \mathcal{H}$ s.t. $\kappa^\dagger(\ell \circ h') = \tilde{\ell} \circ \tilde{h}^*$ , where we ask $h' \neq h^*$, otherwise we would be imposing the trivial condition $\ell \circ h^* = \tilde{\ell} \circ \tilde{h}^* = \kappa^\dagger(\ell \circ h^*)$, i.e. the corruption is harmless w.r.t. the Bayes Risk value. In order to study the possible loss correction, we choose the corrupted optimum as $\tilde{h}^* = h'$ (**A2**).

**Loss corrections**     We now try to formalize how to define a suitable loss for the corrupted learning problem, such that the optimal hypothesis is learned in the clean learning space. The problem setting gives us access to $\ell, \kappa^\dagger$ given by the problem, and $\tilde{h}^*$ chosen by us. We want to find a way to retrieve a suitable $h^*$ for the clean distribution. That means, the loss correction task here is *finding a formulation of $\tilde{\ell}$ that depends on $\ell, \kappa^\dagger$*. An essential preliminary result, for which the proof is given in § S4.1, is

**Lemma 7.** *The feasible factorization of a Markov kernel $\kappa$ is also a valid factorization for its pseudo-inverse $\kappa^\dagger$, both for the full kernel or considering their parameterized versions.*

We then give the correction results (proof in § S4.2), and discuss them.

---

[7] Here $h^*$ is inverted as a function, not as a kernel. That means, $\tilde{\ell}(p, y) = \ell(h^*((\tilde{h}^*)^{-1}(p)), y)$.

**Theorem 8.** *Let $(\ell, \mathcal{H}, P)$ be a clean learning problem and $(\kappa^\dagger(\ell \circ \mathcal{H}), \kappa P)$ its associated corrupted one, not necessarily with a $\circ$-factorized structure. Let $\kappa^\dagger$ be the joint cleaning kernel reversing $\kappa$, such that assumptions **A1** and **A2** hold for the said problems. The factorization of $\kappa^\dagger$ is assumed to be feasible and to have an equality result of the form Eq. (5). We write $\kappa^\dagger(dz, \tilde{z}) = \kappa^X(dx, \cdot)\kappa^Y(dy, \cdot)$, with $(\cdot)$ some feasible parameters. Hence, we can prove the following points:*

1. *When $\kappa^\dagger$ is either $(id_X, \text{S-}Y)$ or $(id_X, \text{2D-}Y)$, we can write the corrected loss as*
$$\tilde{\ell}(h(\tilde{x}), \tilde{y}) = (\kappa^Y \ell)(h(\tilde{x}), \tilde{y}) \quad \forall (\tilde{x}, \tilde{y}) \in \tilde{X} \times \tilde{Y},$$
   *with $\kappa^Y \ell = \kappa_{\tilde{x}}^Y \ell$ for the second case.*

2. *When $\kappa^\dagger$ is $(\text{S-}X, \text{S-}Y)$, $(\text{2D-}X, \text{S-}Y)$ or $(\text{S-}X, \text{2D-}Y)$, we have*
$$\tilde{\ell}(\tilde{x}, \tilde{y}, h) = \mathbb{E}_{\mathsf{u} \sim \kappa^X h(\tilde{x})}[\kappa^Y \ell(\mathsf{u}, \tilde{y})] \quad \forall (\tilde{x}, \tilde{y}) \in \tilde{X} \times \tilde{Y},$$
   *with $\kappa_{\tilde{x}}^X h(\tilde{x})(A) := \kappa^X(h^{-1}(A), \tilde{x})$, $A \subset \mathcal{P}(Y)$ being the push-forward probability measure of $\kappa^X(\cdot, \tilde{x})$ through $h$, $h$ seen as a function. For the cases that involve a 2D corruption, we have $\kappa^Y \ell = \kappa_{\tilde{x}}^Y \ell$ for the former $\kappa^\dagger$ factorization, $\kappa^X h(\tilde{x}) = \kappa_{\tilde{y}}^X h(\tilde{x})$ for the latter.*

3. *When $\kappa^\dagger$ is a $(\text{1D-}X, \text{1D-}Y)$ corruption, we can write the corrected loss as*
$$\tilde{\ell}(\tilde{x}, \tilde{y}, h) = \mathbb{E}_{\mathsf{u} \sim \kappa^X h(\tilde{y})}[\kappa^Y \ell(\mathsf{u}, \tilde{x})] \quad \forall (\tilde{x}, \tilde{y}) \in \tilde{X} \times \tilde{Y},$$
   *with $\kappa_{\tilde{x}}^X h(\tilde{y})(B) := \kappa^X(h^{-1}(B), \tilde{y})$, $B \subset \mathcal{P}(X)$.*

4. *When $\kappa^\dagger$ is a $(\text{2D}, \text{1D})$ corruption, we can write the corrected loss as*
$$\tilde{\ell}(\tilde{x}, \tilde{y}, h) = \mathbb{E}_{\mathsf{u} \sim \kappa^X h(\tilde{y})}[\kappa_{\tilde{x}}^Y \ell(\mathsf{u}, \tilde{y})], \quad \tilde{\ell}(\tilde{x}, \tilde{y}, h) = \mathbb{E}_{\mathsf{u} \sim \kappa_{\tilde{y}}^X h(\tilde{x})}[\kappa^Y \ell(\mathsf{u}, \tilde{x})] \quad \forall (\tilde{x}, \tilde{y}) \in \tilde{X} \times \tilde{Y}.$$
   *for the $(\text{1D-}X, \text{2D-}Y)$, $(\text{2D-}X, \text{1D-}Y)$ respectively.*

When minimized, the corrected losses will by construction give back the hypothesis $\tilde{h}^*$. Since $\ell \circ h^* = \tilde{\ell} \circ \tilde{h}^*$, the learned $\tilde{h}^*$ has on the clean distribution the optimum performance we wanted to achieve with the original loss function $\ell$. Hence, we achieve unbiased learning in the sense of matching scores and in the distributional sense.

The corrections found by the theorem are more complex than the ones defined in previous work [7, 34], i.e., the first part of point 1. In the second part, we characterize the effect of a more "dependent" $Y$ cleaning kernel, i.e. closer to the root in Fig. 1a. When also $\kappa^X$ is non-trivial in the factorization, we have an action on $h$. Then, the corrected functions lie in a larger function space than the usual one, the one of positive, bounded functions $\ell : X \times Y \times \mathcal{H} \to \mathbb{R}^+$.

The result additionally underlines how the cleaning kernel affects the a hypothesis on $X$: it induces a set of "reachable predictions" from $h$ through $\kappa^\dagger$, depending on the outcome of the stochastic process $\kappa^\dagger : \tilde{X} \rightsquigarrow X$. The push-forward probability measures are probabilities on a *set of probabilities*. For instance, in point 2 we have $\kappa_{\tilde{y}}^X h(\tilde{x}) \in \mathcal{P}(\mathcal{P}(Y))$, while for point 3 we have $\kappa_{\tilde{x}}^X h(\tilde{x}) \in \mathcal{P}(\mathcal{P}(X))$.

## 6 Conclusions

We proposed a comprehensive and unified framework for corruption using Markov kernels, systematically studying corruption in three key aspects: classification, consequence, and correction. We established a new taxonomy of corruption, enabling qualitative comparisons between corruption models in terms of the corruption hierarchy. To gain a deeper quantitative understanding of corruption, we analyzed the consequences of different corruptions from an information-theoretic standpoint by proving Data Processing Equalities for Bayes Risk. As a consequence of them, we obtained loss correction formulas that gives us more insights into the effect of corruption on losses.

Throughout the work, we consider data as probability distributions, implicitly assuming that each dataset has an associated probabilistic generative process. We treat corruption as Markov kernels, assuming full access to their actions, and analyze the consequences of corruption through Bayes risks without accounting for sampling or optimization. Bridging the gap between the distributional-level and the sample-level results would be the next step for this study, which requires tailored ad-hoc analyses. Other directions for making this framework more practically usable include developing quantitative methods to compare corruption severity and investigating the effects of optimization algorithms on the analysis.

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
