# Supplementary Material for "A General Framework for Learning under Corruption: Label Noise, Attribute Noise, and Beyond"

## S1 Additional discussions on related work

Here, we detail discussions on the relations with existing paradigms as shown in Tab. 1. As a reminder, we review the commonly used notations. Let $E : Y \rightsquigarrow X$ be an experiment and $F : X \rightsquigarrow Y$ be a posterior kernel. The clean distribution $P$ can be represented either in a discriminative manner as $\pi_X \times F$ or in a generative manner as $\pi_Y \times E$. However, we cannot observe samples drawn from the clean distribution $P$, but observe samples from some corrupted distribution $\tilde{P}$. The corruption is generally represented as $\kappa_{Z\tilde{Z}}$, where the variables $z = (x, y) \in Z$ are referred to as parameters and the differentials $d\tilde{z} = d\tilde{x}d\tilde{y}$ are referred to as corrupted variables. $\delta_{Z\tilde{Z}}$ denotes a kernel induced by the Dirac delta measure from $(Z, \mathcal{Z})$ to $(Z, \mathcal{Z})$.

### S1.1 Simple corruptions

The most well-known and widely studied corruptions in the literature are the simple cases, where the corruption solely acts on the feature space $X$ or the label space $Y$. We discuss examples of the simple corruptions S-$\tilde{X}$ and S-$\tilde{Y}$, as illustrated in Fig. 1a, in the following.

**Attribute noise** The problem of attribute noise concerns errors that are introduced into the observations of attribute X, leaving the labels untouched [30, 31, 4, 19]. Widely studied examples of such errors include erroneous attribute values and missing attribute values. Instead of observing $(X, Y)$, in the first case, one can only observe a distorted version of X, e.g. $(X + N, Y)$ with some independent noise random variable $N \perp\!\!\!\perp X$; in the second case, one's observation of X contains missing values.

Let $\boldsymbol{X} = (x_{ij})_{1 \le i \le n, 1 \le j \le d}$ be the complete input matrix, with $|X| = n$, and $\boldsymbol{M} = (m_{ij})_{1 \le i \le n, 1 \le j \le d}$ be the associated missingness indicator matrix such that $m_{ij} = 1$ if $x_{ij}$ is observed and $m_{ij} = 0$ if $x_{ij}$ is missing. Then the corresponding observed input matrix is $\boldsymbol{X}_o = \boldsymbol{X} \odot \boldsymbol{M}$ and its missing counterpart is $\boldsymbol{X}_m = \boldsymbol{X} - \boldsymbol{X}_o$, where $\odot$ denotes Hadamard product. The missing value mechanisms are further categorized into three types based on their dependencies [5, 6]: [8]

- Missing completely at random (MCAR): the cause of missingness is entirely random, i.e., $p(M \mid X) = p(M)$ does not depend on $X_o$ or $X_m$. This corresponds to having a trivial Markov kernel acting on the clean distribution, $\kappa_{X\tilde{X}} \equiv \mu \in \mathcal{P}(X)$.
- Missing not at random (MNAR): the cause of missingness depends on both observed variables and missing variables, i.e., $p(M \mid X) = p(M \mid X_o, X_m)$. This case corresponds to our non-trivial $\kappa_{X\tilde{X}}$.
- Missing at random (MAR): the cause of missingness depends on observed variables but not on missing variables, i.e., $p(M \mid X) = p(M \mid X_o)$. This case is a sub-case of the non-trivial $\kappa_{X\tilde{X}}$, which is not specifiable by our taxonomy because of the different premises it is built on.

We underline that the conditional distributions of M described above are *not* an equivalent description of our Markov kernels. The missing data case is also known as finite Selection Bias, as discussed in § S2.3, we know there exists a Markov kernel describing this corruption but the definition per se is a *non-stochastic corruption*.

Hence, attribute noise is an example of S-$\tilde{X}$ corruption that can be generally formulated as the corrupted experiment illustrated in the transition diagram $Y \xrightarrow{E} X \xrightarrow{\kappa_{X\tilde{X}}} \tilde{X}$, and the corrupted distribution is given by $\tilde{P} = (\kappa_{X\tilde{X}}\delta_{Y\tilde{Y}}) \circ (\pi_Y \times E)$.

**Class-conditional noise (CCN)** The problem of CCN arises in situations where, instead of observing the clean labels, one can only observe corrupted labels that have been flipped with a label-dependent probability, while the marginal distribution of the instance remains unchanged [5, 34, 7, 19]. CCN is an example of S-$\tilde{Y}$ corruption that can be formulated as a corrupted posterior illustrated in the transition diagram $X \xrightarrow{F} Y \xrightarrow{\kappa_{Y\tilde{Y}}} \tilde{Y}$, and the corrupted distribution is given by $\tilde{P} = (\delta_{X\tilde{X}}\kappa_{Y\tilde{Y}}) \circ (\pi_X \times F)$. For classification tasks, $Y$ and $\tilde{Y}$ are assumed to be finite spaces. Therefore the corruption $\kappa_{Y\tilde{Y}}$ can be represented by a column-stochastic matrix

---

[8]Assume the rows $x_i$, $m_i$ are assigned a joint distribution. and X and M are treated as random variables.

$\boldsymbol{T} = (\rho_{ij})_{1 \leq i \leq |\tilde{Y}|, 1 \leq j \leq |Y|}$ which specifies the probability of the clean label $\mathsf{Y} = j$ being flipped to the corrupted label $\tilde{\mathsf{Y}} = i$, i.e., $\forall i, j$, $\rho_{ij} = p(\tilde{\mathsf{Y}} = i \mid \mathsf{Y} = j)$. The corrupted joint distribution can be rewritten as $\tilde{P} = \sum_Y p(\tilde{\mathsf{Y}} \mid \mathsf{Y})p(\mathsf{Y} \mid \mathsf{X})p(\mathsf{X})$. In the literature, $\boldsymbol{T}$ is known as the noise transition matrix with its elements $\rho_{ij}$ referred to as the noise rates, and is useful for designing loss correction approaches (our results in § 5 significantly generalize existing loss correction results in CCN to our broad class of simple, dependent and combined corruptions) [34]. Prior to the proposal of the CCN model, early studies primarily focused on a symmetric subcase of $\boldsymbol{T}$ in binary classification, known as random classification noise (RCN) [32, 33, 12]. Note that in RCN, the output of the corruption $\kappa_{\tilde{Y}}$ remains constant w.r.t. its parameters. Recently, some variants of CCN have been further developed, for example, in Ishida et al. [13, 14], complementary labels that can be modeled via a symmetric $\boldsymbol{T}$ whose diagonal elements are all equal to zero are studied.

## S1.2 Dependent corruptions

Although simple corruptions have been well studied and understood, more complexities arise in dependent cases, yet they receive relatively less attention and understanding. We discuss examples of the dependent corruptions 1D-$\tilde{X}$, 1D-$\tilde{Y}$, 2D-$\tilde{X}$ and 2D-$\tilde{Y}$, as illustrated in Fig. 1a, in the following.

**Style transfer**  Style transfer refers to the process of migrating the artistic style of a given image to the content of another image [35, 36]. The primary objective is to recreate the second image with the designated style of the first image. In recent developments, it has also been applied to audio signals [37]. If we represent the style of the first image by $\mathsf{Y}$, and the second image and the reconstructed image as $\mathsf{X}$ and $\tilde{\mathsf{X}}$ respectively, style transfer serves as an illustrative example of 1D-$\tilde{X}$ "corruption". Note that the aim here is to *learn how to corrupt* instead of learning in the presence of corruption. We mention this connection because our framework can also be used also with different purposes, but underline that our BR results are not applicable to this case. The process of style transfer can be formulated as a corrupted posterior illustrated in the transition diagram $X \overset{F}{\rightsquigarrow} Y \overset{\kappa_{Y\tilde{X}}}{\rightsquigarrow} \tilde{X}$ , and the corrupted distribution is given by $\tilde{P} = (\kappa_{Y\tilde{X}}\delta_{Y\tilde{Y}}) \circ (\pi_X \times F)$.

**Adversarial noise**  In contrast to additive random attribute noise, adversarial noise is specifically crafted by adversaries for each instance with the intent of changing the model's prediction of the correct label [38, 39, 40, 41, 42]. Such adversarial examples raise significant security concerns as they can be utilized to attack machine learning systems, even in scenarios where the adversary has no access to the underlying model. The adversarial noise is an example of 2D-$\tilde{X}$ corruption that can be formulated as a corrupted experiment illustrated in the transition diagram $Y \overset{E}{\rightsquigarrow} X \overset{\kappa_{XY\tilde{X}}}{\underset{\kappa_{XY\tilde{X}}}{\rightsquigarrow}} \tilde{X}$ , and the corrupted distribution is given by $\tilde{P} = (\kappa_{XY\tilde{X}}\delta_{Y\tilde{Y}}) \circ (\pi_Y \times E)$.

**Instance-dependent noise (IDN)**  As a counterpart to CCN, the problem of IDN arises in situations where, instead of observing the clean labels, one can only observe corrupted labels that have been flipped with an instance-dependent (but not label-dependent) probability [18, 8]. It is a special case of the ILN noise model, which we will describe later. IDN is an example of 1D-$\tilde{Y}$ corruption that can be formulated as a corrupted experiment illustrated in the transition diagram $Y \overset{E}{\rightsquigarrow} X \overset{\kappa_{X\tilde{Y}}}{\rightsquigarrow} \tilde{Y}$ , and the corrupted distribution is given by $\tilde{P} = (\delta_{X\tilde{X}}\kappa_{X\tilde{Y}}) \circ (\pi_Y \times E)$.

**Instance- and label-dependent noise (ILN)**  ILN is the most general label noise model, which arises in situations where, instead of observing clean labels, one can only observe corrupted labels that have been flipped with an instance- and label-dependent probability [8, 43, 44, 45]. ILN is an example of 2D-$\tilde{Y}$ corruption that can be formulated as a corrupted posterior illustrated in the transition diagram $X \overset{F}{\rightsquigarrow} Y \overset{\kappa_{XY\tilde{Y}}}{\underset{\kappa_{XY\tilde{Y}}}{\rightsquigarrow}} \tilde{Y}$ , and the corrupted distribution is given by $\tilde{P} = (\delta_{X\tilde{X}}\kappa_{XY\tilde{Y}}) \circ (\pi_X \times F)$. Compared to the matrix representation $\boldsymbol{T}$ of the CCN corruption $\kappa_{Y\tilde{Y}}$, the ILN corruption $\kappa_{XY\tilde{Y}}$ can be represented by a matrix-valued function of the instance $\boldsymbol{T}(x) = (\rho_{ij}(x))_{1 \leq i \leq |\tilde{Y}|, 1 \leq j \leq |Y|}$ which specifies the probability that the instance $\mathsf{X} = x$ with the clean label $\mathsf{Y} = j$ being flipped to the corrupted label $\tilde{\mathsf{Y}} = i$, i.e., $\forall i, j$, $\rho_{ij}(x) = p(\tilde{\mathsf{Y}} = i \mid \mathsf{Y} = j, \mathsf{X} = x)$. Some subcases of ILN have

also been studied in the literature, for example, the boundary-consistent noise, which considers a label flip probability based on a score function of the instance and label. The score aligns with the underlying class-posterior probability function, resulting in instances closer to the optimal decision boundary having a higher chance of its label being flipped [23].

### S1.3 Combined corruptions

Given the simple and dependent corruptions, we can combine them to generate 2-parameter joint corruptions, i.e., $\kappa_{Z\tilde{Z}} : X \times Y \rightsquigarrow \tilde{X} \times \tilde{Y}$. Below, we discuss some examples of combined noise models illustrated in Fig. 1b.

**Combined simple noise** The simplest combined corruption is the combined simple noise, where the observations of attribute $X$ are subject to some errors and the observed labels $Y$ are flipped with a label-dependent probability [19]. Combined simple noise is an example of $(S\text{-}\tilde{X}, S\text{-}\tilde{Y})$ corruption that can be formulated as a corrupted experiment illustrated in the transition diagram $\tilde{Y} \overset{\kappa_{Y\tilde{Y}}}{\rightsquigarrow} Y \overset{E}{\rightsquigarrow} X \overset{\kappa_{X\tilde{X}}}{\rightsquigarrow} \tilde{X}$ , and the corrupted distribution is given by $\tilde{P} = (\kappa_{X\tilde{X}}\kappa_{Y\tilde{Y}}) \circ (\pi_Y \times E)$.

**Target shift** In the literature, target shift refers to the situation where the prior probability $p(Y)$ is changed while the conditional distribution $p(X \mid Y)$ remains invariant across training and test domains [46, 47, 48, 49]. The definition is established by assuming certain invariance from a generative perspective of the learning problem, that is, considering it as a corruption of the experiment according to $P = \pi_Y \times E$. However, when examining the learning problem from a discriminative perspective, the change in $p(Y)$ may cause changes in both $p(X)$ and $p(Y \mid X)$ due to the Bayes rule. Existing frameworks for the categorization of target shift do not capture these implications, as they are based on the notion of invariance from a single perspective of the $E$ direction. In contrast, our framework categorizes corruptions based on their dependencies and therefore is advantageous by offering dual perspectives from both the $E$ and $F$ directions. Specifically, target shift is an example of $(1D\text{-}\tilde{X}, 2D\text{-}\tilde{Y})$ corruption and can be formulated either as a corrupted experiment illustrated in the transition diagram $\tilde{X} \overset{\kappa_{Y\tilde{X}}}{\rightsquigarrow} Y \underset{\kappa_{XY\tilde{Y}}}{\overset{E}{\rightsquigarrow}} X \overset{\kappa_{XY\tilde{Y}}}{\rightsquigarrow} \tilde{Y}$ , or as a corrupted posterior illustrated in the transition diagram $\tilde{Y} \underset{\kappa_{XY\tilde{Y}}}{\overset{\kappa_{XY\tilde{Y}}}{\rightsquigarrow}} X \overset{F}{\rightsquigarrow} Y \overset{\kappa_{Y\tilde{X}}}{\rightsquigarrow} \tilde{X}$ . The corrupted distribution is given by $\tilde{P} = (\kappa_{Y\tilde{X}}\kappa_{XY\tilde{Y}}) \circ (\pi_Y \times E)$ or $\tilde{P} = (\kappa_{Y\tilde{X}}\kappa_{XY\tilde{Y}}) \circ (\pi_X \times F)$.

**Mutually contaminated distributions (MCD)** The problem of MCD arises in binary classification situations where, instead of observing samples from the clean class-conditional distributions $p(X \mid Y = \pm 1)$, one can only observe samples from corrupted class-conditional distributions $\tilde{p}(X \mid Y = \pm 1)$, with

$$\begin{pmatrix} \tilde{p}(X \mid Y = +1) \\ \tilde{p}(X \mid Y = -1) \end{pmatrix} = \begin{pmatrix} 1 - \alpha & \alpha \\ \beta & 1 - \beta \end{pmatrix} \begin{pmatrix} p(X \mid Y = +1) \\ p(X \mid Y = -1) \end{pmatrix}$$

as described in [6, 50, 51]. The coefficients $\alpha$ and $\beta$ are defined as the fraction of data points having a flipped label, given that the true one was respectively $+1$ or $-1$.

In comparison, CCN corrupts the class-posterior probability $p(Y \mid X)$ while MCD corrupts the class-conditional distribution $p(X \mid Y)$; consequently, the marginal distribution of $p(X)$ remains unchanged in CCN but may be changed in MCD. Therefore $\alpha$ and $\beta$ in MCD are not the noise rates $\rho_{12}$ and $\rho_{21}$ in CCN, however, they are shown to be related by an invertible transformation [6]. In other words, CCN is shown to be a subcase of the MCD, but what else is included in the MCD model is not explored. Therefore, here we model MCD as $(2D\text{-}\tilde{X}, S\text{-}\tilde{Y})$ corruption, which can be formulated as a corrupted experiment illustrated in the transition diagram $\tilde{Y} \overset{\kappa_{Y\tilde{Y}}}{\rightsquigarrow} Y \underset{\kappa_{XY\tilde{X}}}{\overset{E}{\rightsquigarrow}} X \overset{\kappa_{XY\tilde{X}}}{\rightsquigarrow} \tilde{X}$ , and the corrupted distribution is given by $\tilde{P} = (\kappa_{XY\tilde{X}}\kappa_{Y\tilde{Y}}) \circ (\pi_Y \times E)$.

**Covariate shift** In the literature, covariate shift refers to the situation where the marginal distribution $p(X)$ is changed while the class-posterior probability $p(Y \mid X)$ remains invariant across training and test domains [3, 52, 53, 54]. Similarly to target shift, the definition is established by assuming certain

invariance from a discriminative perspective of the learning problem. However, when examining the learning problem from a generative perspective, the change in $p(\mathsf{X})$ may cause changes in $p(\mathsf{Y})$ and $p(\mathsf{X} \mid \mathsf{Y})$ due to the Bayes rule. Covariate shift in its general definition is an example of $(2\text{D-}\tilde{X}, 1\text{D-}\tilde{Y})$ corruption and can be formulated either as a corrupted posterior illustrated in the transition diagram

$$\tilde{Y} \overset{\kappa_{X\tilde{Y}}}{\rightsquigarrow} X \xrightarrow[\underset{\kappa_{XY\tilde{X}}}{}]{F} Y \overset{\kappa_{XY\tilde{X}}}{\rightsquigarrow} \tilde{X} \text{ , or as a corrupted experiment illustrated in the transition}$$

diagram

$$\tilde{X} \overset{\kappa_{XY\tilde{X}}}{\underset{\kappa_{XY\tilde{X}}}{\rightsquigarrow}} Y \xrightarrow{E} X \overset{\kappa_{X\tilde{Y}}}{\rightsquigarrow} \tilde{Y} \text{ . The corrupted distribution is given by } \tilde{P} = (\kappa_{XY\tilde{X}} \kappa_{X\tilde{Y}}) \circ (\pi_X \times F)$$

or $\tilde{P} = (\kappa_{XY\tilde{X}} \kappa_{XY\tilde{Y}}) \circ (\pi_Y \times E)$.

**Generalized target shift**  In the literature, generalized target shift refers to the situation where the prior probability $p(\mathsf{Y})$ and the conditional distribution $p(\mathsf{X} \mid \mathsf{Y})$ both change across training and test domains, however, with some invariance assumptions in the latent space [55, 56, 57]. Generalized target shift is an example of $(2\text{D-}\tilde{X}, 2\text{D-}\tilde{Y})$ corruption that can be formulated as a

corrupted experiment illustrated in the transition diagram $\tilde{Y} \overset{\kappa_{XY\tilde{Y}}}{\rightsquigarrow} Y \xrightarrow{E} X \overset{\kappa_{XY\tilde{X}}}{\rightsquigarrow} \tilde{X}$ , and the

corrupted distribution is given by $\tilde{P} = (\kappa_{XY\tilde{X}} \kappa_{XY\tilde{Y}}) \circ (\pi_Y \times E)$. Note that simpler scenarios can also result in a generalized target shift, however, it is important to avoid degenerating to the simple S-$\tilde{X}$ corruption, as it would violate the requirement of corrupting the label distribution.

**Concept drift**  Concept drift refers to the situation where $\tilde{p}(\mathsf{Y} \mid \mathsf{X}) \neq p(\mathsf{Y} \mid \mathsf{X})$ [17]. As in the case of generalized target shift, this case can be associated with every corruption in our framework, so the most general correspondence is the $(2\text{D-}\tilde{X}, 2\text{D-}\tilde{Y})$ joint Markov kernel.

# S2  Appendix for "A general framework for corruption", Section 3

## S2.1  The superposition operation

We further describe the superposition operation between kernels, also known as "parallel combination" [27] or Kronecker product when in finite spaces, by specifying the action of the resulting kernel on functions and measures.

**Definition S1.** *Let $\kappa_1$ be a Markov kernel from $(X, \mathcal{X})$ to $(Y, \mathcal{Y})$ and $\kappa_2$ be a Markov kernel from $(Z, \mathcal{Z})$ to $(W, \mathcal{W})$. Hence, the **superposition** of the two is a kernel $\kappa_1\kappa_2$ from $(X \times Z, \mathcal{X} \times \mathcal{Z})$ to $(Y \times W, \mathcal{Y} \times \mathcal{W})$ such that:*

$$(\kappa_1\kappa_2)f(x,z) = \int_{Y \times W} (\kappa_1\kappa_2)(x, z, dydw)\, f(y, w)$$
$$= \int_Y \kappa_1(x, dy) \int_W \kappa_2(z, dw)\, f(y, w)\ ,$$

*for every $f$ positive $\mathcal{Y} \times \mathcal{W}$-measurable, or equivalently*

$$\mu(\kappa_1\kappa_2)(B) = \int_B \int_{X \times Z} (\kappa_1\kappa_2)(x, z, dydw)\, \mu(dxdz)$$
$$= \int_B \int_X \kappa_1(x, dy) \int_Z \kappa_2(z, dw)\, \mu(dxdz)\ ,$$

*for every measure $\mu$ on $(X \times Z, \mathcal{X} \times \mathcal{Z})$, $B \in \mathcal{Y} \times \mathcal{W}$.*

Both the operators are well defined, as we can rewrite them

$$(\kappa_1\kappa_2)f(x, z) = \kappa_1(\kappa_2 f(y, w)) = \kappa_1 \hat{f}(y, z)\ ,$$
$$\mu(\kappa_1\kappa_2)(B_Y \times B_W) = \kappa_1(\mu\kappa_2)(B_Y \times B_W) = \hat{\mu}\kappa_1(B_Y \times B_W)\ .$$

Hence we are just iteratively applying the standard kernel-induced operators to a parameterized function or partially to a joint measure.

When dealing with finite spaces, Markov kernels are column-stochastic matrices. The superposition operation is then equal to the *Kronecker product*, between two matrices,

$$\kappa_1 \kappa_2 := \kappa_1 \otimes \kappa_2 = \begin{pmatrix} (\kappa_1)_{11}\kappa_2 & \dots & (\kappa_1)_{1n}\kappa_2 \\ \vdots & \ddots & \vdots \\ (\kappa_1)_{m1}\kappa_2 & \dots & (\kappa_1)_{mn}\kappa_2 \end{pmatrix}$$

with $\kappa_1$ being a $|Y| \times |X|$ matrix and $\kappa_2$ a $|W| \times |Z|$ matrix.

## S2.2 The Bayesian inversion theorem

In this section, we present some existing results coming from category theory applied to Bayesian learning, which allows us to define and use the inverse kernel as introduced in the main text. The background knowledge required for following this section is rather different from that of the other sections. However, even if readers choose not to delve into the specific details, they can still comprehend our results by only referring to the notions in Def. S4.

In Dahlqvist et al. [26], they address the question of Markov kernel inversion through the lens of category theory.[9] They investigate how and when the (weak) inversion is defined, both directly on the category of measurable spaces and indirectly by considering the associated Markov linear operator (Markov transition [42]). We only focus on illustrating the first result, given the focus on Markov kernels we had in the paper. We will use category theory terminology, and then connect it to our probabilistic vocabulary.

The first step is the construction of the Krn category, similar to our notion of space of Markov kernels $\mathcal{M}(X, Y)$ but with an equivalence relation acting on it. They start by considering Polish metric spaces, the category Pol with continuous mappings, a subcase of which are the closed sets $X \subseteq \mathbb{R}^d$ equipped with the usual topology. The category of measurable spaces considered for defining kernels, Mes, is the one induced by a functor $\mathcal{B} : \text{Pol} \rightarrow \text{Mes}$, i.e. all the measurable spaces with the same underlying set of a Polish space but equipped by the Borel $\sigma$-algebra and interpreting continuous mapping as measurable ones. We call these spaces *standard Borel spaces*, and use them as the building block of the Krn category.

The category Mes is embedded by a functor $F$ into the Kleisli category of G, a monad over Mes representing probability distributions over some set. The functor $F$ acts *identically* on sets and *maps* measurable functions $f : X \rightarrow Y$ to Kleisli arrows $F(f) = \delta_Y \circ f$. This means, in more familiar terms, that we build a trivial kernel $\delta_Y(f^{-1}(dy))$, i.e. the image measure of the dirac delta through $f$. It further induces the category $1 \downarrow F$ of probabilities $p : 1 \rightarrow \text{G}X$ and trivial morphisms $f : (X, p) \rightarrow_\delta (Y, q)$ as degenerate arrows $F(f)$ s.t. $q = F(f) \circ_\text{G} p = \text{G}(f)(p)$, where $\circ_\text{G}$ corresponds our $\circ$ combination between a kernel and a distribution. In other words, $F(f)$ induces a measure-preserving map, so $1 \downarrow F$ includes all measure-preserving maps induced by *degenerate* arrows. When the arrows used are not degenerate, we obtain the supercategory $1 \downarrow \mathcal{Kl}$, with the same objects. We denote arrows in this category as $f : (X, p) \rightarrow (Y, q)$. Notice that the kernels included in the category $1 \downarrow \mathcal{Kl}$ are what we would call $\mathcal{M}(X, Y)$, where $X$ has marginal $p$ and $Y$ has marginal $q$.

This last category includes Markov kernels as we have defined them in this paper. They are considered as *typed kernels*, i.e. their definition is tied to a fixed input and a fixed output (probabilities) instead of being characterized for every input probability and every reachable output. This remark is crucial for understanding our notion of exhaustiveness – we will later underline why.

Markov kernels cannot be inverted as they are, because of their non-singularity. They characterize it with their Lemma 3, proving that for a kernel $f : (X, p) \rightarrow (Y, q)$ there are *p-negligibly many points jumping to q-negligible sets*. Once the non-singularity is understood, we can define an equivalence relation that, when acting on $1 \downarrow \mathcal{Kl}$, allows a well-posed definition of the inverse kernel.

**Definition S2.** *For all objects $(X, p), (Y, q)$, $R_{(X,p),(Y,q)}$ is the smallest equivalence relation on $Hom_{1 \downarrow \mathcal{Kl}}(X, Y)$ such that*

$$(f, f') \in R_{(X,p),(Y,q)} \quad \Leftrightarrow \quad f = f' \ p - a.s.$$

In Lemma 4, they prove $R$ to be a congruence relation on $1 \downarrow \mathcal{Kl}$. This congruence relation allows us to define the quotient category, with the proper morphisms.

---

[9] For a general overview, see Mac Lane [41].

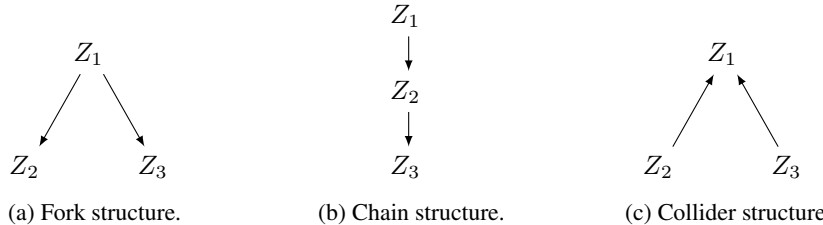

(a) Fork structure.   (b) Chain structure.   (c) Collider structure.

Figure S1: Possible non-degenerate relations among three probability spaces.

**Definition S3.** *The category* Krn *is the quotient category* $(1 \downarrow \mathcal{K}\ell)/R$ .

Having defined the category, we have to build the functions that are going to constitute the weak inversion operator, i.e. a bijection between $\mathrm{Hom}_{\mathtt{Krn}}((X,p),(Y,q))$ and $\mathrm{Hom}_{\mathtt{Krn}}((Y,q),(X,p))$. They are two mapping between the Krn category and the space of couplings associated to $(X,p),(Y,q)$. The first is equivalent to our $\times$ kernel operation, applied to a kernel (i.e. conditional probability) and a probability, and is formally written as

$$\alpha_Y^X : \mathrm{Hom}_{\mathtt{Krn}}((X,p),(Y,q)) \to \Gamma((X,p),(Y,q)) \quad \text{s.t.} \quad \alpha_Y^X(f)(B_X \times B_Y) := \int_{x \in B_X} f(x)(B_Y)dp \,,$$

with $\Gamma((X,p),(Y,q)) \subset \mathtt{G}(X,Y)$ the typed couplings associated to the marginals $(X,p),(Y,q)$.

The second is defined as its inverse operation, and it is decomposing a joint probability along a fixed marginal distribution, i.e.,

$$D_Y^X : \Gamma((X,p),(Y,q)) \to \mathrm{Hom}_{\mathtt{Krn}}((X,p),(Y,q)) \quad \text{s.t.} \quad D_Y^X(\gamma) := G(\pi_Y) \circ \pi_X^\dagger \,,$$

such that

$$\gamma(B_X \times B_Y) := \int_{x \in B_X} D_Y^X(\gamma)(x)(B_Y)dp \,,$$

with $(\cdot)^\dagger$: adjoint operator. Being one the inverse of the other, they are both obviously bijective and proving the one-to-one correspondence between typed kernels and typed couplings.

Hence, we formally define the pseudo-inverse as in the following:

**Definition S4.** *The inverse of a typed kernel* $\kappa$ *from* $(X_1,p_1)$ *to* $(X_2,p_2)$, *given by* $\kappa^\dagger \circ \kappa :=$ $D_X^Y \circ \mathtt{G}(\pi_2 \times \pi_1) \circ \alpha_Y^X(\kappa)$ *with* $\mathtt{G}(\pi_2 \times \pi_1) : \Gamma((X_1,p_1),(X_2,p_2)) \to \Gamma((X_2,p_2),(X_1,p_1))$ *being the permutation map, is defined as*

1. $\kappa^\dagger : (X_2,p_2) \to (X_1,p_1) \in$ Krn *when* $\kappa$ *is seen as element of* Krn, *such that* $\kappa^\dagger \circ \kappa = \delta_{X_1}$ *and* $\kappa \circ \kappa^\dagger = \delta_{X_2}$ ;

2. $\kappa^\dagger : X_2 \rightsquigarrow X_1 \in \mathcal{M}(X_2,X_1)$ *when* $\kappa$ *is seen as element of* $\mathcal{M}(X_1,X_2)$, *such that* $\kappa^\dagger \circ \kappa =_R \delta_{X_1}$ *and* $\kappa \circ \kappa^\dagger =_R \delta_{X_2}$ .

## S2.3 Exhaustiveness of the taxonomy

In the previous section, we define the operations $\alpha$ and $D$ for typed kernels, which are one the inverse of the other by construction. They are the operations representing the bijection between the space of Markov kernels typed for $p,q$ and the space of couplings with marginals $p,q$. Hence, they are proving that *for each couple of probability spaces, there exists a Markov kernel sending one into the other corresponding to a possible associated coupling.*

This means that every pairwise stochastic corruption in the supervised learning setting is described by our taxonomy. Other possibilities are, having more than two spaces involved in the corruption process and having a deterministic mapping describing the corruption process as it has been defined. We discuss them in the following, providing examples.

**Stochastic corruption for more than two spaces** When in the presence of three probability spaces, we have only two possible corruption configurations. We represent them in Fig. S1, where arrows represent non-trivial Markov kernels. We remark that we do not consider the triangular structures

as in Fig. S1a and c with the spaces $Z_2, Z_3$ coupled in some way, otherwise they would just be considered as a single (product) probability space, i.e. a pairwise corruption.

The most simple case is Fig. S1b, in which the spaces influence each other in a chain fashion. This is a clear subcase of our framework as we can integrate $Z_2$ by considering $\kappa_{Z_1 Z_3} \coloneqq \kappa_3 \circ \kappa_2 \circ \kappa_1$. We then obtain a pairwise corruption $\kappa_{Z_1 Z_3}$, but we would pay the price of losing information about the role of the 'latent' corruption process. To have a complete idea of how the chained corruption works, we can additionally study it as an iterative process and analyze its single components. This entire reasoning is true for a number of spaces $Z_i, i \in [n]$ with $n > 3$, and well models several settings for dynamical learning, e.g. online corrupted learning or concept drift over time [58, 61, 59].

The second option is, they act as per the diagrams in Fig. S1a and Fig. S1c, i.e. a triangular structure. In particular, case (a) reflects assumptions made in settings combining data from different domains [29, 7, 62], where we get to observe different data distributions obtained from the same clean one. They can be seen, in our framework, as a pairwise dependence between $Z_1 \times Z_2$ and $Z_3$, or $Z_1$ and $Z_2 \times Z_3$. However, this formulation assumes some coupling on $Z_2, Z_3$, more complex than our originally assumed corruption. For now, we do not investigate the consequences of this gap as changes of the corruption effect, leaving it for future investigation. A similar idea can be stated for $n > 2$ spaces in the Cartesian product space, and for combinations of fork structures with fork structures via superposition.

**Corruptions via deterministic mappings**  We now want to give examples of how corruption processes can be not stochastic. From the previous section, we know that even if there is no direct way of modeling the specific corruption process with a Markov kernel, there exists a Markov kernel representing some coupling between two distributions. We do not define any method to find the *best* Markov corruption corresponding to a deterministic one, since it depends on the specific task one is considering.

The first relevant example is the one of **Selection Bias**. Even if being a widely studied, common case of corruption, we show here that when considered with its classical formulation we cannot directly find a Markov kernel corresponding to it.

We start by introducing the Selection Bias type corruption, as done in [14]. It is characterized here as a distributional corruption, unlike other cases in which only the selection variable is modeled. We consider a target, clean distribution and a source, corrupted one from which we aim to learn. They are defined on the same set $Z \subseteq \mathbb{R}^d$ and Borel $\sigma$-algebra $\mathcal{Z}$. We define it as:

1. Support condition:

$$\tilde{P} \ll P \iff \exists! \; \alpha = \frac{d\tilde{P}}{dP} \; a.s., \; \alpha \in L^1 \;,$$

where we can equivalently say $\mu - a.s., \mu \coloneqq 0.5 * (\tilde{P} + P)$, or $P - a.s.$;

2. Selection condition:

$$\|\alpha\|_\infty < +\infty \;.$$

The Support condition is equivalent to:

$$\tilde{P}(A) = \int_A \alpha(z) P(dz) \; \forall A \;.$$

Comparing it with a Markov kernel action on the input probability $P$, we get the condition

$$\int_A \int_Z \kappa(z, d\tilde{z}) P(dz) = \int_A \alpha(z) P(dz) \quad \forall A \;.$$

A guess that satisfies our requirement is $\kappa(z, d\tilde{z}) \coloneqq \delta_z(d\tilde{z})\alpha(z)$, which is a transition kernel, i.e. a family of positive measures parameterized by $z$, *but not* a Markov kernel unless $\alpha \equiv 1$. This kernel is defined such that $P$ is corrupted into $\tilde{P}$, but it does not preserve mass for every probability measure. *Is this the only possible guess?*

Assuming the existence of a transition kernel $\hat{\kappa}(z, d\tilde{z}) \neq \delta_z(d\tilde{z})\alpha(z)$, possibly Markov, implies

$$\int_A \int_Z \hat{\kappa}(z, d\tilde{z}) P(dz) = \int_A \tilde{P}(dz) = \int_A \alpha(z) P(dz) \; \forall A \;.$$

729  We then can define a measure $\hat{\mu}(d\tilde{z}) := \int_Z \hat{\kappa}(z, d\tilde{z})P(dz)$. This measure $\hat{\mu}$ is almost surely equal to
730  $\tilde{P}$ by definition, w.r.t. a reference measure $\mu_1$. The same is true for $\tilde{P}$ and $\alpha P$ w.r.t. $\mu_2$. Hence $\hat{\mu}(dz)$
731  is equal to $\alpha(z)P(dz)$ w.r.t. to $\mu$, $\mu_1 \ll \mu$ and $\mu_2 \ll \mu$.

732  Since the same argument can be repeated for the kernel $\kappa$, the two kernels are forced to be equal
733  $\mu$-almost surely. That because, for two measures with the same value on every set, their Radon-
734  Nikodym derivatives are the same almost everywhere w.r.t. a finite[10] reference measure. Hence,
735  Selection Bias cannot be directly represented as a Markov kernel if we impose it to be acting *exactly*
736  as the weak derivative $\alpha$.

737  Another relevant example is the one of **Markov kernel reconstruction** $R$, as introduced in [7]. They
738  are considered in finite space settings and are defined as the *left inverse of the stochastic matrix*
739  *representing the Markov kernel*. It is underlined by the authors that the $R$ of the Markov kernel is
740  not necessarily a Markov kernel; in fact, it is not even ensured to be a matrix with positive entries.
741  A reconstruction $R$ is then sending a corrupted probability $\tilde{P}$ into the original clean probability $P$
742  without being a stochastic mapping.

# S3  Appendix for "Consequences of corruption in supervised learning", Section 4: Proofs

745  We restate for clarity all the assumptions underlying the proofs.

746  **A1** We assume the loss function to be bounded in order to avoid problems when applying Fubini-
747  Tonelli's theorem.
748  **A2** We define the set $\ell \circ \mathcal{H} := \{ (x, y) \mapsto \ell(h(x), y) \mid h \in \mathcal{H} \}$.
749  **A3** When minimizing the risk for the corrupted distribution $\tilde{P}$, we assume that $f^* \in$
750  $\arg\min_f \mathbb{E}_{\tilde{P}}[f(X, Y)]$ belongs to the minimization space $\ell \circ \mathcal{H}$.

751  Theorems 3, 4 are here proved by means of two Lemmas on the dependent noise combined with
752  identical simple noise.

753  **Lemma S5** (BR under X corruption). *Let $(\ell, \mathcal{H}, P)$ be a learning problem with the input space $X$*
754  *and output space $Y$. Let $E : Y \rightsquigarrow X$ be an experiment, $\kappa^{\tilde{X}} \in \{\kappa_{X\tilde{X}}, \kappa_{YX\tilde{X}}\}$ be the corruption*
755  *on $X$ with at most 2 parameters, then we can form the corrupted experiment as per the transition*
756  *diagram* $Y \underset{\kappa^{\tilde{X}}}{\overset{E}{\rightsquigarrow}} X \overset{\kappa^{\tilde{X}}}{\rightsquigarrow} \tilde{X}$ *and obtain*

$$\mathbb{E}_{Y \sim \pi_Y} CBR_{\ell \circ \mathcal{H}}(\kappa^{\tilde{X}} E_Y) = \mathbb{E}_{Y \sim \pi_Y} CBR_{\kappa^{\tilde{X}}(\ell \circ \mathcal{H})}(E_Y) \,.$$

757  *Moreover, if $\kappa^{\tilde{X}} = \kappa_{X\tilde{X}}$, we have*

$$BR_{\ell \circ \mathcal{H}}(\pi_Y \times \kappa^{\tilde{X}} E) = BR_{\kappa^{\tilde{X}}(\ell \circ \mathcal{H})}(\pi_Y \times E) \,.$$

758  *Proof.* Assume the full corruption $\kappa$ has an associated kernel

$$\kappa(x, y, d\tilde{x}d\tilde{y}) := (\kappa^{\tilde{X}}\delta)(x, y, d\tilde{x}d\tilde{y}) = \kappa_y^{\tilde{X}}(x, d\tilde{x})\delta_y(d\tilde{y}), \tag{S1}$$

759  Let $E_y(dx) := E(y, dx)$ and $A \in \tilde{\mathcal{X}} \times \tilde{\mathcal{Y}}$, we have

$$\begin{aligned}
\tilde{P}(A) &= \sum_Y \int_A \int_X \kappa(x, y, d\tilde{x}d\tilde{y}) \ P(dxdy) \\
&= \sum_Y \int_A \int_X \kappa_y^{\tilde{X}}(x, d\tilde{x})\delta_y(d\tilde{y}) \ E_y(dx)\pi_y \\
&= \sum_Y \int_A (\kappa^{\tilde{X}}E)_y(d\tilde{x}) \ \delta_y(d\tilde{y})\pi_y \\
&= \int_A \tilde{E}_{\tilde{y}}(d\tilde{x})\pi_{\tilde{y}},
\end{aligned}$$

---

[10]It is enough to ask "finite on all balls", see [63], Theorem 5.8.8.

then we can write

$$\mathbb{E}_{\tilde{Y}\sim\pi_{\tilde{Y}}}CBR_{\ell\circ\mathcal{H}}(\kappa^{\tilde{X}}E_{\tilde{Y}}) = \sum_{\tilde{Y}}\pi_{\tilde{y}}\inf_{f\in\ell\circ\mathcal{H}}\int_{\tilde{X}}f(\tilde{x},\tilde{y})\,\tilde{E}_{\tilde{y}}(d\tilde{x})$$

$$= \sum_{Y,\tilde{Y}}\delta_y(d\tilde{y})\pi_y\inf_{f\in\ell\circ\mathcal{H}}\int_{\tilde{X}X}f(\tilde{x},\tilde{y})\,\kappa_y^{\tilde{X}}(x,d\tilde{x})E_y(dx)$$

$$= \sum_{Y}\pi_y\inf_{f\in\ell\circ\mathcal{H}}\int_{\tilde{X}}\delta f(\tilde{x},y)\int_{X}\kappa_y^{\tilde{X}}(x,d\tilde{x})E_y(dx)$$

$$= \sum_{Y}\pi_y\inf_{f\in\ell\circ\mathcal{H}}\int_{X}E_y(dx)\,(\kappa_y^{\tilde{X}}\delta f)(x,y) \qquad (S2)$$

$$= \sum_{Y}\pi_y\inf_{f\in\kappa^{\tilde{X}}(\ell\circ\mathcal{H})}\int_{X}E_y(dx)\,f(x,y)$$

$$= \mathbb{E}_{Y\sim\pi_Y}CBR_{\kappa^{\tilde{X}}(\ell\circ\mathcal{H})}(E_Y). \qquad (S3)$$

Since the X corruption $\kappa^{\tilde{X}}$ has an identity mapping on Y, $\mathbb{E}_{\tilde{Y}\sim\pi_{\tilde{Y}}}[\cdot] = \mathbb{E}_{Y\sim\pi_Y}[\cdot]$ and we obtain

$$\mathbb{E}_{Y\sim\pi_Y}CBR_{\ell\circ\mathcal{H}}(\kappa^{\tilde{X}}E_Y) = \mathbb{E}_{Y\sim\pi_Y}CBR_{\kappa^{\tilde{X}}(\ell\circ\mathcal{H})}(E_Y).$$

If $\kappa^{\tilde{X}} = \kappa_{X\tilde{X}}$, then the associated kernel (S1) takes the simple form $\kappa^{\tilde{X}}(x,d\tilde{x})$ and the above equations from (S2) become

$$BR_{\ell\circ\mathcal{H}}(\pi_Y\times\kappa^{\tilde{X}}E) = \inf_{f\in\ell\circ\mathcal{H}}\sum_{Y}\pi_y\int_{X}E_y(dx)\,(\kappa^{\tilde{X}}f)(x,y)$$

$$= \inf_{f\in\kappa^{\tilde{X}}(\ell\circ\mathcal{H})}\sum_{Y}\pi_y\int_{X}E_y(dx)\,f(x,y)$$

$$= BR_{\kappa^{\tilde{X}}(\ell\circ\mathcal{H})}(\pi_Y\times E).$$

$\qquad\qquad\qquad\qquad\qquad\qquad\qquad\qquad\qquad\qquad\qquad\qquad\qquad\qquad\qquad\qquad\square$

**Theorem** (BR under (S-$\tilde{X}$, S-$\tilde{Y}$), (2D-$\tilde{X}$, S-$\tilde{Y}$) joint corruption, Theorem 3). *Let $(\ell,\mathcal{H},P)$ be a learning problem, $E : Y \rightsquigarrow X$ an experiment and $\kappa^{\tilde{X}} \in \{\kappa_{X\tilde{X}}, \kappa_{YX\tilde{X}}\}$ a corruption as in Lemma S5. Let $\kappa_{Y\tilde{Y}}$ be a simple corruption on Y. Then we can form the corrupted experiment as per the transition diagram $\tilde{Y} \xrightarrow{\kappa_{Y\tilde{Y}}} Y \xrightarrow[\kappa^{\tilde{X}}]{E} X \xrightarrow{\kappa^{\tilde{X}}} \tilde{X}$ and obtain*

$$\mathbb{E}_{\tilde{Y}\sim\kappa_{Y\tilde{Y}}\pi_Y}CBR_{\ell\circ\mathcal{H}}(\kappa^{\tilde{X}}E_{\tilde{Y}}) = \mathbb{E}_{Y\sim\pi_Y}CBR_{\kappa^{\tilde{X}}(\kappa_{Y\tilde{Y}}\ell\circ\mathcal{H})}(E_Y)$$

*Proof.* We assume the full corruption $\kappa$ has an associated kernel

$$\kappa(x,y,d\tilde{x}d\tilde{y}) \coloneqq (\kappa^{\tilde{X}}\kappa_{Y\tilde{Y}})(x,y,d\tilde{x}d\tilde{y}) = \kappa_y^{\tilde{X}}(x,d\tilde{x})\kappa_y^{Y\tilde{Y}}(d\tilde{y})\,.$$

With this corruption formulation, we can replicate the proof of Lemma S5 up to (S2) by simply plugging in $\kappa_y^{Y\tilde{Y}}(d\tilde{y})$ instead of $\delta_y(d\tilde{y})$. Therefore, we obtain the thesis. $\qquad\square$

**Lemma S6** (BR under Y corruption). *Let $E : Y \rightsquigarrow X$ and $F : X \rightsquigarrow Y$ be an experiment and a posterior on it, and $\kappa^{\tilde{Y}} \in \{\kappa_{Y\tilde{Y}}, \kappa_{XY\tilde{Y}}\}$ be the corruption on Y with at most 2 parameters, then we can form the corrupted posterior as per the transition diagram $X \xrightarrow[\kappa^{\tilde{Y}}]{F} Y \xrightarrow{\kappa^{\tilde{Y}}} \tilde{Y}$ and obtain*

$$\mathbb{E}_{X\sim\pi_X}CBR_{\ell\circ\mathcal{H}}(\kappa^{\tilde{Y}}F_X) = \mathbb{E}_{X\sim\pi_X}CBR_{\kappa^{\tilde{Y}}\ell\circ\mathcal{H}}(F_X)\,.$$

*Moreover, if $\kappa^{\tilde{Y}} = \kappa_{Y\tilde{Y}}$, the equation simplifies as*

$$BR_{\ell\circ\mathcal{H}}(\pi_X\times\kappa^{\tilde{Y}}F) = BR_{\kappa^{\tilde{Y}}(\ell)\circ\mathcal{H}}(\pi_X\times F)\,.$$

*Equivalently, we can form the corrupted experiment as per the transition diagram*

$Y \overset{E}{\rightsquigarrow} \tilde{X} \overset{\kappa}{\rightsquigarrow} \tilde{Y}$ *and obtain*

$$BR_{\ell \circ \mathcal{H}}(\kappa(\pi_Y \times E)) = BR_{\kappa(\ell \circ \mathcal{H})}(\pi_Y \times E) \, ,$$

*where $\kappa = \kappa^{\tilde{Y}} \delta_x$, the combination of $\kappa^{\tilde{Y}}$ with the identity kernel on $X$.*

*Proof.* Assume the full corruption $\kappa$ has an associated kernel

$$\kappa(x, y, d\tilde{x} d\tilde{y}) := (\kappa^{\tilde{Y}} \delta)(x, y, d\tilde{x} d\tilde{y}) = \kappa_x^{\tilde{Y}}(y, d\tilde{y}) \delta_x(d\tilde{x}) \, . \tag{S4}$$

Let $F_x(dy) := F(x, dy)$ and $A \in \tilde{\mathcal{X}} \times \tilde{\mathcal{Y}}$, we have

$$\begin{aligned}
\tilde{P}(A) &= \sum_Y \int_A \int_X \kappa(x, y, d\tilde{x} d\tilde{y}) \, P(dx dy) \\
&= \sum_Y \int_A \int_X \kappa_x^{\tilde{Y}}(y, d\tilde{y}) \delta_x(d\tilde{x}) \, F_x(dy) \pi_x \\
&= \int_A \int_X (\kappa^{\tilde{Y}} F)_x(d\tilde{y}) \, \delta_x(d\tilde{x}) \pi_x \\
&= \int_A \tilde{F}_{\tilde{x}}(d\tilde{y}) \pi_{\tilde{x}},
\end{aligned}$$

then we can write

$$\begin{aligned}
\mathbb{E}_{\tilde{X} \sim \pi_{\tilde{X}}} CBR_{\ell \circ \mathcal{H}}(\kappa^{\tilde{Y}} F_{\tilde{X}}) &= \int_{\tilde{X}} \pi_{\tilde{x}} \inf_{f \in \ell \circ \mathcal{H}} \sum_{\tilde{Y}} f(\tilde{x}, \tilde{y}) \, \tilde{F}_{\tilde{x}}(d\tilde{y}) \\
&= \int_{\tilde{X} \times X} \delta_x(d\tilde{x}) \pi_x \inf_{f \in \ell \circ \mathcal{H}} \sum_{Y, \tilde{Y}} f(\tilde{x}, \tilde{y}) \, \kappa_x^{\tilde{Y}}(y, d\tilde{y}) F_x(dy) \\
&= \int_X \pi_x \inf_{f \in \ell \circ \mathcal{H}} \sum_{Y, \tilde{Y}} \delta f(x, \tilde{y}) \, \kappa_x^{\tilde{Y}}(y, d\tilde{y}) F_x(dy) \\
&= \int_X \pi_x \inf_{f \in \ell \circ \mathcal{H}} \sum_Y F_x(dy) (\kappa_x^{\tilde{Y}} \delta f)(x, y) \tag{S5} \\
&= \int_X \pi_x \inf_{f \in \kappa^{\tilde{Y}}(\ell \circ \mathcal{H})} \sum_Y F_x(dy) f(x, y) \\
&= \mathbb{E}_{X \sim \pi_X} CBR_{(\kappa^{\tilde{Y}} \ell \circ \mathcal{H})}(F_X). \tag{S6}
\end{aligned}$$

Since the Y corruption $\kappa^{\tilde{Y}}$ has an identity mapping on $X$, $\mathbb{E}_{\tilde{X} \sim \pi_{\tilde{X}}}[\cdot] = \mathbb{E}_{X \sim \pi_X}[\cdot]$ and we obtain

$$\mathbb{E}_{X \sim \pi_X} CBR_{\ell \circ \mathcal{H}}(\kappa^{\tilde{Y}} F_X) = \mathbb{E}_{X \sim \pi_X} CBR_{\kappa^{\tilde{Y}}(\ell \circ \mathcal{H})}(F_X).$$

If $\kappa^{\tilde{Y}} = \kappa_{Y\tilde{Y}}$, then the associated kernel (S4) takes the simple form $\kappa^{\tilde{Y}}(y, d\tilde{y})$ and the above equations from (S5) become

$$\begin{aligned}
BR_{\ell \circ \mathcal{H}}(\pi_X \times \kappa^{\tilde{Y}} F) &= \inf_{f \in \ell \circ \mathcal{H}} \int_X \pi_x \sum_Y F_x(dy) (\kappa^{\tilde{Y}} f)(x, y) \\
&= \inf_{f \in \kappa^{\tilde{Y}}(\ell \circ \mathcal{H})} \int_X \pi_x \sum_Y F_x(dy) f(x, y) \\
&= BR_{\kappa^{\tilde{Y}}(\ell \circ \mathcal{H})}(\pi_X \times F).
\end{aligned}$$

In this case, reminding that $h(x, y) = (h(x), id(y))$, we have

$$BR_{\kappa^{\tilde{Y}}(\ell \circ \mathcal{H})}(\pi_X \times F) = BR_{\kappa^{\tilde{Y}}(\ell) \circ \mathcal{H}}(\pi_X \times F) \, .$$

Similarly, the results can also be expressed in terms of $E$ using the generic corruption formulation:

$$BR_{\ell \circ \mathcal{H}}(\kappa \pi_Y \times E)) = BR_{\ell \circ \mathcal{H}}(\kappa P) = \inf_{f \in \ell \circ \mathcal{H}} \int_{\tilde{X}} \sum_{\tilde{Y}} f(\tilde{x}, \tilde{y}) \, \kappa P(d\tilde{x} d\tilde{y})$$

$$= \inf_{f \in \ell \circ \mathcal{H}} \int_{\tilde{X} \times X} \sum_{Y, \tilde{Y}} f(\tilde{x}, \tilde{y}) \kappa(x, y, d\tilde{x} d\tilde{y}) P(dx dy)$$

$$= \inf_{f \in \ell \circ \mathcal{H}} \int_X \sum_Y \kappa f(x, y) P(dx dy)$$

$$= BR_{\kappa(\ell \circ \mathcal{H})}(\pi_Y \times E) \tag{S7}$$

Note that the last result in (S7), even if fitting in the comparison of experiments literature [20], does not give us any new insights since it is not based on the corruption decomposition formula in (S4). We provide (S7) here for completeness. $\qquad \square$

**Theorem** (BR under (S-$\tilde{X}$, S-$\tilde{Y}$), (S-$\tilde{X}$, 2D-$\tilde{Y}$) joint corruption, Theorem 4). *Let* $(\ell, \mathcal{H}, P)$ *be a learning problem,* $F : X \rightsquigarrow Y$ *a posterior and* $\kappa^{\tilde{Y}} \in \{\kappa_{Y\tilde{Y}}, \kappa_{XY\tilde{Y}}\}$ *a corruption as in Lemma S6. Let* $\kappa_{X\tilde{X}}$ *be a simple corruption on* $X$*. Then we can form the corrupted experiment as per the transition diagram* $\tilde{X} \xrightarrow{\kappa_{X\tilde{X}}} X \xrightarrow{F} Y \xrightarrow{\kappa^{\tilde{Y}}} \tilde{Y}$ *and obtain*

$$\mathbb{E}_{\tilde{X} \sim \kappa_{X\tilde{X}} \pi_X} CBR_{\ell \circ \mathcal{H}}(\kappa^{\tilde{Y}} F_{\tilde{X}}) = \mathbb{E}_{X \sim \pi_X} CBR_{\kappa_{X\tilde{X}}(\kappa^{\tilde{Y}} \ell \circ \mathcal{H})}(F_X) . \tag{S8}$$

*Proof.* We assume the full corruption $\kappa$ has an associated kernel

$$\kappa(x, y, d\tilde{x} d\tilde{y}) \coloneqq (\kappa^{\tilde{Y}} \kappa^{X\tilde{X}})(x, y, d\tilde{x} d\tilde{y}) = \kappa_x^{\tilde{Y}}(y, d\tilde{y}) \kappa_x^{X\tilde{X}}(d\tilde{x}) .$$

With this corruption formulation, we can replicate the proof of Lemma S6 up to (S5) by simply plugging in $\kappa_x^{X\tilde{X}}(d\tilde{x})$ instead of $\delta_x(d\tilde{x})$. Therefore, we obtain the thesis. $\qquad \square$

**Remark S7.** *When using the continuous notation for $Y$ we do so for simplicity and homogeneity. Notice that all its associated kernel are actually (parameterized) squared matrices, hence transposable. So in Theorem 4 and the associated Lemma, the operator acting on the function is actually the transpose of the corruption matrix.*

$$\sum_{\tilde{y}} C_{\tilde{y}y}(x) \ell_{\tilde{y}}(h(x)) = \sum_{\tilde{y}} C_{y\tilde{y}}^T(x) \ell_{\tilde{y}}(h(x)) = (\ell_y \circ h)_x^*(x) .$$

**Theorem** (BR under (1D, 2D) joint corruption, Theorem 5). *Let* $(\ell, \mathcal{H}, P)$ *be a learning problem,* $E : Y \rightsquigarrow X$ *and* $F : X \rightsquigarrow Y$ *be an experiment and a posterior on it.*

*1. Let* $\kappa_{Y\tilde{X}}$ *be a corruption on* $X$ *and* $\kappa_{XY\tilde{Y}}$ *be a corruption on* $Y$*, then we can form the jointly corrupted experiment as per the transition diagram* $\tilde{X} \xrightarrow{\kappa_{Y\tilde{X}}} Y \xrightarrow{E} X \xrightarrow{\kappa_{XY\tilde{Y}}} \tilde{Y}$ *and obtain*

$$BR_{\ell \circ \mathcal{H}}[\kappa_{Y\tilde{X}}(\pi_Y \times \kappa_{XY\tilde{Y}} E)] = \mathbb{E}_{Y \sim \pi_Y} CBR_{\kappa_{Y\tilde{X}}(\kappa_{XY\tilde{Y}} \ell \circ \mathcal{H})}(E_Y) . \tag{S9}$$

*2. Let* $\kappa_{X\tilde{Y}}$ *be a corruption on* $Y$ *and* $\kappa_{XY\tilde{X}}$ *be a corruption on* $X$*, then we can form the jointly corrupted posterior as per the transition diagram* $\tilde{Y} \xrightarrow{\kappa_{X\tilde{Y}}} X \xrightarrow{F} Y \xrightarrow{\kappa_{XY\tilde{X}}} \tilde{X}$ *and obtain*

$$BR_{\ell \circ \mathcal{H}}[\kappa_{X\tilde{Y}}(\pi_X \times \kappa_{XY\tilde{X}} F)] = \mathbb{E}_{X \sim \pi_X} CBR_{\kappa_{XY\tilde{X}}(\kappa_{X\tilde{Y}} \ell \circ \mathcal{H})}(F_X) . \tag{S10}$$

*Proof.* For proving point (1), assume the full corruption $\kappa$ has an associated kernel

$$\kappa(x, y, d\tilde{x} d\tilde{y}) \coloneqq (\kappa_{Y\tilde{X}} \kappa_{XY\tilde{Y}})(x, y, d\tilde{x} d\tilde{y}) = \kappa^{Y\tilde{X}}(y, d\tilde{x}) \kappa_y^{XY\tilde{Y}}(x, d\tilde{y}) .$$

803    Let $E_y(dx) := E(y, dx)$ and $A \in \tilde{\mathcal{X}} \times \tilde{\mathcal{Y}}$, we have

$$
\begin{aligned}
\tilde{P}(A) &= \sum_Y \int_A \int_X \kappa(x, y, d\tilde{x}d\tilde{y}) \, P(dxdy) \\
&= \sum_Y \int_A \int_X \kappa^{Y\tilde{X}}(y, d\tilde{x}) \kappa_y^{XY\tilde{Y}}(x, d\tilde{y}) \, E_y(dx)\pi(dy) \\
&= \int_A \sum_Y \kappa^{Y\tilde{X}}(y, d\tilde{x})(\pi_Y \times (\kappa^{XY\tilde{Y}}E))(dy, d\tilde{y}) \\
&= \int_A \kappa^{Y\tilde{X}}(\pi_Y \times \kappa^{XY\tilde{Y}}E)(d\tilde{x}, d\tilde{y}) \,,
\end{aligned}
$$

804    which is less interpretable as a corruption action if compared to the previous theorems, since the
805    effect on $E$ and $\pi_Y$ cannot be totally distinguished. However, we can still write

$$
\begin{aligned}
\mathbb{E}_{\tilde{Y}\sim\tilde{\pi}_{\tilde{Y}}} CBR_{\ell\circ\mathcal{H}}(\kappa_{Y\tilde{X}}\kappa_{XY\tilde{Y}}E) &= \sum_{\tilde{Y}} \inf_{f\in\ell\circ\mathcal{H}} \int_{\tilde{X}} f(\tilde{x}, \tilde{y}) \, \tilde{P}(d\tilde{x}d\tilde{y}) \\
&= \sum_Y \int_{\tilde{X}} \kappa^{Y\tilde{X}}(y, d\tilde{x})\pi_y \inf_{f\in\ell\circ\mathcal{H}} \sum_{\tilde{Y}} \int_X f(\tilde{x}, \tilde{y}) \, \kappa_x^{XY\tilde{Y}}(y, d\tilde{y}) E_y(dx) \\
&= \sum_Y \int_{\tilde{X}} \kappa^{Y\tilde{X}}(y, d\tilde{x})\pi_y \inf_{\ell\circ h\in\ell\circ\mathcal{H}} \int_X (\kappa_x^{XY\tilde{Y}}\ell)(h(\tilde{x}), y) \, E_y(dx) \\
&= \sum_Y \pi_y \inf_{f\in\ell\circ\mathcal{H}} \int_X \kappa^{Y\tilde{X}}[(\kappa_x^{XY\tilde{Y}}\ell)_y \circ h](y) \, E_y(dx) \\
&= \sum_Y \pi_y \inf_{f\in\kappa_{Y\tilde{X}}(\kappa_{XY\tilde{Y}}\ell\circ\mathcal{H})} \int_X E_y(dx) f(x, y, h) \\
&= \mathbb{E}_{Y\sim\pi_Y} CBR_{\kappa_{Y\tilde{X}}(\kappa_{XY\tilde{Y}}\ell\circ\mathcal{H})}(E) \,,
\end{aligned}
$$

806    with $f(x, y, h) := \kappa^{Y\tilde{X}}[(\kappa_x^{XY\tilde{Y}}\ell)_y \circ h](y)$. In particular, notice that $\kappa^{XY\tilde{Y}}$ acts only on $\ell$, while
807    $\kappa_{Y\tilde{X}}$ acts on both $\ell$ and $h$, which forces us to use $f(x, y, h)$ instead of $f(x, y)$.

808    For point (2), assume the full corruption $\kappa$ has an associated kernel

$$
\kappa(x, y, d\tilde{x}d\tilde{y}) := (\kappa_{X\tilde{Y}}\kappa_{XY\tilde{X}})(x, y, d\tilde{x}d\tilde{y}) = \kappa^{X\tilde{Y}}(x, d\tilde{y})\kappa_x^{XY\tilde{X}}(y, d\tilde{x}) \,.
$$

809    Let $F_x(dy) := F(x, dy)$ and $A \in \tilde{\mathcal{X}} \times \tilde{\mathcal{Y}}$, we have

$$
\begin{aligned}
\tilde{P}(A) &= \sum_Y \int_A \int_X \kappa(x, y, d\tilde{x}d\tilde{y}) \, P(dxdy) \\
&= \sum_Y \int_A \int_X \kappa^{X\tilde{Y}}(x, d\tilde{y})\kappa_x^{XY\tilde{X}}(y, d\tilde{x}) \, F_x(dy)\pi(dx) \\
&= \int_A \int_X \kappa^{X\tilde{Y}}(x, d\tilde{y})(\pi_X \times \kappa^{XY\tilde{X}}F)(d\tilde{x}, dx) \\
&= \int_A \kappa^{X\tilde{Y}}(\pi_X \times \kappa^{XY\tilde{X}}F)(d\tilde{x}, d\tilde{y}) \,,
\end{aligned}
$$

810    Hence we can repeat a similar argument for the $F$ case and find a minimization space of functions
811    $f(x, y, h) := \kappa^{XY\tilde{X}}[(\kappa^{X\tilde{Y}}\ell)_x \circ h]_y(x)$. Thus, we obtain the thesis. $\qquad\square$

812    **Corollary** (BR under (1D, 1D) joint corruption, Corollary 6)**.** *Let $(\ell, \mathcal{H}, P)$ be a learning problem,*
813    *$E : Y \rightsquigarrow X$ and $F : X \rightsquigarrow Y$ be an experiment and a posterior on it. Let $\kappa_{Y\tilde{X}}$ be a corruption on $X$*
814    *and $\kappa_{X\tilde{Y}}$ be a corruption on $Y$, then we can form the jointly corrupted experiment as per the transition*
815    *diagram* $\tilde{X} \xrightarrow{\kappa_{Y\tilde{X}}} Y \xrightarrow{E} X \xrightarrow{\kappa_{X\tilde{Y}}} \tilde{Y}$ *or equivalently* $\tilde{Y} \xrightarrow{\kappa_{X\tilde{Y}}} X \xrightarrow{F} Y \xrightarrow{\kappa_{Y\tilde{X}}} \tilde{X}$. *We*
816    *obtain*

$$
BR_{\ell\circ\mathcal{H}}[\kappa_{Y\tilde{X}}(\pi_Y \times \kappa_{X\tilde{Y}}E)] = BR_{\kappa_{Y\tilde{X}}(\kappa_{X\tilde{Y}}\ell\circ\mathcal{H})}(\pi_Y \times E) \,,
$$

817    *or equivalently*

$$
BR_{\ell\circ\mathcal{H}}[\kappa_{X\tilde{Y}}(\pi_X \times \kappa_{Y\tilde{X}}F)] = BR_{\kappa_{Y\tilde{X}}(\kappa_{X\tilde{Y}}\ell\circ\mathcal{H})}(\pi_X \times F) \,.
$$

*Proof.* We assume the full corruption $\kappa$ has an associated kernel

$$\kappa(x,y,d\tilde{x}d\tilde{y}) := (\kappa^{Y\tilde{X}}\kappa^{X\tilde{Y}})(x,y,d\tilde{x}d\tilde{y}) = \kappa^{Y\tilde{X}}(y,d\tilde{x})\kappa^{X\tilde{Y}}(x,d\tilde{y}) \ .$$

With this corruption formulation, we can replicate the proof of Theorem 5 by simply plugging in $\kappa^{X\tilde{Y}}(x,d\tilde{y})$ instead of $\kappa^{XY\tilde{Y}}(x,y,d\tilde{y})$ in the first point, and by simply plugging in $\kappa^{Y\tilde{X}}(y,d\tilde{x})$ instead of $\kappa^{XY\tilde{X}}(x,y,d\tilde{x})$ in the second point. We then in both cases obtain functions $f(x,y,h) := \kappa^{Y\tilde{X}}[(\kappa^{X\tilde{Y}}\ell)_x \circ h](y)$, i.e. comparing a point $x$ with a kernel on $\mathcal{P}(X)$ parameterized by $y$.

After getting the identities w.r.t. CBRs, we can further take the $\inf$ operator out of the outside expectations and obtain identities w.r.t. BRs, as the kernels $\kappa^{Y\tilde{X}}$ and $\kappa^{X\tilde{Y}}$ are not parameterized by $x$ or $y$ anymore. Therefore, we obtain the thesis. $\qquad\square$

**Analysis of Bayes Risk under (2D-$\tilde{X}$, 2D-$\tilde{Y}$)** Let $(\ell,\mathcal{H},P)$ be a learning problem, $E : Y \rightsquigarrow X$ and $F : X \rightsquigarrow Y$ be an experiment and a posterior on it. Let $\kappa_{XY\tilde{X}}$ be the corruption on $X$ and $\kappa_{XY\tilde{Y}}$ be the corruption on $Y$. Then we can form the jointly corrupted experiment as per the transition diagram $\tilde{Y} \overset{\kappa_{XY\tilde{X}}}{\rightsquigarrow} Y \overset{E}{\rightsquigarrow} X \overset{\kappa_{XY\tilde{X}}}{\rightsquigarrow} \tilde{X}$. Hence, the full corruption $\kappa$ has an associated kernel

$$\kappa(x,y,d\tilde{x}d\tilde{y}) := (\kappa_{XY\tilde{X}}\kappa_{XY\tilde{Y}})(x,y,d\tilde{x}d\tilde{y}) = \kappa_y^{\tilde{X}}(x,d\tilde{x})\kappa_x^{\tilde{Y}}(y,d\tilde{y}) \ .$$

Let $\tilde{P} = \kappa P$ and $A \in \tilde{\mathcal{X}} \times \tilde{\mathcal{Y}}$, we have

$$\tilde{P}(A) = \sum_Y \int_A \int_X \kappa(x,y,d\tilde{x}d\tilde{y}) \ P(dxdy)$$

$$= \sum_Y \int_A \int_X \kappa_y^{\tilde{X}}(x,d\tilde{x})\kappa_x^{\tilde{Y}}(y,d\tilde{y}) \ E_y(dx)\pi_y \ .$$

This is not further decomposable as an action on the experiment and an action on the prior, given the double dependence of both factors in the kernel.

A similar observation can be done for the posterior kernel, i.e. considering the equivalent transition diagram $\tilde{X} \overset{\kappa_{XY\tilde{X}}}{\rightsquigarrow} X \overset{F}{\rightsquigarrow} Y \overset{\kappa_{XY\tilde{Y}}}{\rightsquigarrow} \tilde{Y}$. Then, we can only write

$$BR_{\ell\circ\mathcal{H}}(\kappa(\pi_Y \times E)) = \inf_{f\in\ell\circ\mathcal{H}} \mathbb{E}_{(\tilde{X},\tilde{Y})\sim\tilde{P}} f(\tilde{X},\tilde{Y})$$

$$= \inf_{f\in\ell\circ\mathcal{H}} \sum_{\tilde{Y}} \tilde{\pi}_{\tilde{y}} \int_{\tilde{X}} f(\tilde{x},\tilde{y}) \ \tilde{E}_{\tilde{y}}(d\tilde{x})$$

$$= \inf_{f\in\ell\circ\mathcal{H}} \sum_{Y,\tilde{Y}} \pi_y \int_{\tilde{X}\times X} \kappa_x^{\tilde{Y}}(y,d\tilde{y})\kappa_y^{\tilde{X}}(x,d\tilde{x})f(\tilde{x},\tilde{y})E_y(dx) \ ,$$

which is, from a Bayesian Risk point of view, equivalent to the non-decomposed joint corruption effect.

# S4 Appendix for "Corruption-corrected learning", Section 5

In this section, we give the proofs of the results used in § 5.

## S4.1 Proof of Lemma 7: Factorization of the pseudo-inverse

**Lemma.** *The feasible factorization of a Markov transition $\kappa$ is also a valid factorization for its pseudo-inverse $\kappa^\dagger$, both for the full transition or considering their parameterized versions.*

*Proof.* Let's consider a Markov kernel $\kappa : X_1 \times Y_1 \to X_2 \times Y_2$ . Also assume that $\kappa = \kappa_X \kappa_Y$, i.e. factorizes by superposition with $\kappa_{(\cdot)} : X_1 \times Y_1 \to (\cdot)_2, (\cdot)_2 \in \{X_2, Y_2\}$.

Supposing that $\kappa_X^\dagger \kappa_Y^\dagger$ is a pseudo-inverse, we can write by using the definition of pseudo-inverse Def. S4:

$$\delta_{x_1', y_1'}(dx_1 dy_1) = \int_{X_2 \times Y_2} \kappa(dx_2 dy_2, x_1', y_1')\kappa^\dagger(dx_1 dy_1, x_2, y_2)$$
$$= \int_{X_2 \times Y_2} (\kappa_X)_{y_1'}(dx_2, x_1')(\kappa_Y)_{x_1'}(dy_2, y_1')(\kappa_X^\dagger)_{y_1'}(dx_1, x_2)(\kappa_Y^\dagger)_{x_1'}(dy_1, x_1', y_2)$$

which shows that $\kappa^\dagger =_R \kappa_X^\dagger \kappa_Y^\dagger$, and proves the Lemma for the parameterized case. Being the regular case a subcase, we obtain the thesis. $\qquad\square$

## S4.2 Proof for Theorem 8

In addition to the assumptions **A1** $-$ **A3** stated for proving the BR theorems (§ S3), we assume here:

**A4** We will assume the existence of an invertible function $\tilde{h}^* \in \mathcal{H}$;

**A5** We ask the corrupted optimum to satisfy the equality $\kappa^\dagger(\ell \circ \tilde{h}^*) = \tilde{\ell} \circ \tilde{h}^*$.

**Theorem.** *Let* $(\ell, \mathcal{H}, P)$ *be a clean learning problem and* $(\kappa^\dagger(\ell \circ \mathcal{H}), \kappa P)$ *its associated corrupted one, not necessarily with a $\circ$-factorized structure. Let* $\kappa^\dagger$ *be the joint cleaning kernel reversing $\kappa$, such that assumptions **A4** and **A5** hold for the said problems. The factorization of $\kappa^\dagger$ is assumed to be feasible and to have an equality result of the form Eq. (5). We write* $\kappa^\dagger(dz, \tilde{z}) = \kappa^X(dx, \cdot)\kappa^Y(dy, \cdot)$, *with $(\cdot)$ some feasible parameters. Hence, we can prove the following points:*

1. *When $\kappa^\dagger$ is either $(id_X, S\text{-}Y)$ or $(id_X, 2D\text{-}Y)$, we can write the corrected loss as*

$$\tilde{\ell}(h(\tilde{x}), \tilde{y}) = (\kappa^Y \ell)(h(\tilde{x}), \tilde{y}) \quad \forall (\tilde{x}, \tilde{y}) \in \tilde{X} \times \tilde{Y} ,$$

*with $\kappa^Y \ell = \kappa_{\tilde{x}}^Y \ell$ for the second case.*

2. *When $\kappa^\dagger$ is $(S\text{-}X, S\text{-}Y)$, $(2D\text{-}X, S\text{-}Y)$ or $(S\text{-}X, 2D\text{-}Y)$, we have*

$$\tilde{\ell}(\tilde{x}, \tilde{y}, h) = \mathbb{E}_{\mathsf{u} \sim \kappa^X h(\tilde{x})}[\kappa^Y \ell(\mathsf{u}, \tilde{y})] \quad \forall (\tilde{x}, \tilde{y}) \in \tilde{X} \times \tilde{Y} ,$$

*with $\kappa_{\tilde{x}}^X h(\tilde{x})(A) := \kappa^X(h^{-1}(A), \tilde{x})$, $A \subset \mathcal{P}(Y)$ being the push-forward probability measure of $\kappa^X(\cdot, \tilde{x})$ through $h$, $h$ seen as a function. For the cases that involve a 2D corruption, we have $\kappa^Y \ell = \kappa_{\tilde{x}}^Y \ell$ for the former $\kappa^\dagger$ factorization, $\kappa^X h(\tilde{x}) = \kappa_{\tilde{y}}^X h(\tilde{x})$ for the latter.*

3. *When $\kappa^\dagger$ is a $(1D\text{-}X, 1D\text{-}Y)$ corruption, we can write the corrected loss as*

$$\tilde{\ell}(\tilde{x}, \tilde{y}, h) = \mathbb{E}_{\mathsf{u} \sim \kappa^X h(\tilde{y})}[\kappa^Y \ell(\mathsf{u}, \tilde{x})] \quad \forall (\tilde{x}, \tilde{y}) \in \tilde{X} \times \tilde{Y} ,$$

*with $\kappa_{\tilde{x}}^X h(\tilde{y})(B) := \kappa^X(h^{-1}(B), \tilde{y})$, $B \subset \mathcal{P}(X)$.*

4. *When $\kappa^\dagger$ is a $(2D, 1D)$ corruption, we can write the corrected loss as*

$$\tilde{\ell}(\tilde{x}, \tilde{y}, h) = \mathbb{E}_{\mathsf{u} \sim \kappa^X h(\tilde{y})}[\kappa_{\tilde{x}}^Y \ell(\mathsf{u}, \tilde{y})] , \quad \tilde{\ell}(\tilde{x}, \tilde{y}, h) = \mathbb{E}_{\mathsf{u} \sim \kappa_{\tilde{y}}^X h(\tilde{x})}[\kappa^Y \ell(\mathsf{u}, \tilde{x})] \quad \forall (\tilde{x}, \tilde{y}) \in \tilde{X} \times \tilde{Y} .$$

*for the $(1D\text{-}X, 2D\text{-}Y)$, $(2D\text{-}X, 1D\text{-}Y)$ respectively.*

*Proof.* Given the assumptions **A4**, **A5**, we can write:

$$\tilde{\ell}(\tilde{h}^*(\tilde{x}), \tilde{y}) = \sum_Y \int_X \ell(\tilde{h}^*(x), y) \, \kappa^\dagger(dxdy, \tilde{x}, \tilde{y}) .$$

We now look at all the feasible corruption combinations in Fig. 1b; given Lemma S4.1, are sure that there factorizations on $\kappa$ are also valid for $\kappa^\dagger$. Hence, we can consider the single point of the theorem being sure that they cover every possible $\kappa$ case having an Bayes Risk equality result.

Consider the $\kappa^\dagger$ from point (1), i.e. $\kappa^\dagger$ is either $(id_X, \text{S-}Y)$ or $(id_X, \text{2D-}Y)$. They act on $\ell \circ h$ as

$$\tilde{\ell}(\tilde{h}^*(\tilde{x}), \tilde{y}) = \sum_Y \int_X \ell(\tilde{h}^*(x), y) \, \delta(dx, \tilde{x}) \, \kappa^Y(dy, \tilde{x}, \tilde{y})$$

$$= \int_X (\kappa^Y \ell)_{\tilde{x}}(h(x), \tilde{y}) \, \delta(dx, \tilde{x})$$

$$= (\kappa^Y \ell)_{\tilde{x}}(h(\tilde{x}), y) \ .$$

Hence, the case $\kappa^Y(dy, \tilde{x}, \tilde{y}) = \kappa_{\tilde{x}}^Y(dy, \tilde{x}, \tilde{y})$ and its subcase $\kappa^Y(dy, \tilde{y})$ combined with an identity kernel on $X$ do not change the hypothesis function.

For the more complex cases in point (2), $\kappa^X(dx, \tilde{x}) \neq \delta_x(dx)$, we have:

$$\tilde{\ell}(\tilde{h}^*(\tilde{x}), \tilde{y}) = \sum_Y \int_X \ell(\tilde{h}^*(x), y) \, \kappa^X(dx, \tilde{x}) \, \kappa^Y(dy, \tilde{x}, \tilde{y})$$

$$= \sum_Y \int_{\tilde{h}^*(X)} \ell(u, y) \, \kappa^X((\tilde{h}^*)^{-1}(du), \tilde{x}) \, \kappa^Y(dy, \tilde{x}, \tilde{y})$$

$$= \int_{\tilde{h}^*(X)} (\kappa^Y \ell)_{\tilde{x}}(u, \tilde{y}) \, \kappa^X((\tilde{h}^*)^{-1}(du), \tilde{x}) \ , \tag{S11}$$

where $u = u(dy) \in \mathcal{P}(Y)$. The following equality for the expectation of $u$, the image measure of $\kappa^\dagger$ through $\tilde{h}^*$, and the kernel chain composition holds:

$$\mathbb{E}_{\kappa^X((\tilde{h}^*)^{-1}(\cdot), \tilde{x})}[u] = \int_{\tilde{h}*(X)} u \, \kappa^X((\tilde{h}^*)^{-1}(du), \tilde{x}) = \kappa^X \circ \tilde{h}^*(\tilde{x}) \in \mathcal{P}(Y) \ ,$$

that can be verified easily by recalling the alternative definition of $\mathcal{H}$ as a subset of $\mathcal{M}(X, Y)$ and using the definition of $\kappa^\dagger \circ \tilde{h}^*$. We remark that $\kappa^X((\tilde{h}^*)^{-1}(du), \tilde{x})$ is then a probability in $\mathcal{P}(\mathcal{P}(Y))$. Hence we can rewrite Eq. (S11) as

$$\tilde{\ell}(\tilde{h}^*(\tilde{x}), \tilde{y}) = \int_{\mathcal{P}(Y)} (\kappa^Y \ell)_{\tilde{x}}(u, \tilde{y}) \, \kappa^X((\tilde{h}^*)^{-1}(du), \tilde{x})$$

$$= \mathbb{E}_{\kappa^X(\tilde{h}^*)^{-1}(\cdot), \tilde{x})}[(\kappa^Y \ell)_{\tilde{x}}(u, \tilde{y})] \ ,$$

with $\kappa^X$ having support included in $\tilde{h}^*(X)$.

As for more dependent corruptions of $X$, i.e. $\kappa^X(dx, \tilde{x}, \tilde{y})$, the action on the hypothesis will be dependent from $\tilde{y}$, so

$$\tilde{\ell}(\tilde{h}^*(\tilde{x}), \tilde{y}) = \mathbb{E}_{\kappa_{\tilde{y}}^X((\tilde{h}^*)^{-1}(\cdot), \tilde{x})}[(\kappa^Y \ell)_{\tilde{x}}(u, \tilde{y})] \ .$$

where only the simple $Y$ noise can be considered, given the missing result for the BR equality in the (D2, D2) joint corruption case.

As for the cases involving the 1D with 1D or 2D, i.e. points (3) and (4), we follow the same procedure by using the action formula of dependent corruptions as described in the proof of Theorem 5, and obtain the thesis. $\qquad\square$

## S5   Table: Actions and consequences of corruption

Table S1: Action of corruption on joint distribution and minimization sets, written using **P1,P2,P3**. We keep the continuous notation also for $Y$ for ease of notation. Here we refer to the Bayes Risk results as the informal equivalence $(\kappa(\tilde{\ell} \circ \tilde{H}), P) \equiv_{BR} (\tilde{\ell} \circ \tilde{H}, \kappa P)$, and call $\tilde{\ell} \circ \tilde{H}$ the minimization set associated to $\kappa P$ (differently from the § S3 Theorems, where we used $\ell \circ H$).

| | Integral representation | Graphical model | Bayes Risk results |
|---|---|---|---|
| S-$\tilde{X}$, S-$\tilde{Y}$ | $\tilde{P} = (\kappa_{Y\tilde{Y}} * \kappa_{X\tilde{X}}) \circ (\pi_Y \times E)$ $= \sum_Y \kappa(d\tilde{y}, y)\pi_Y(dy)\int_X \kappa(d\tilde{x}, x)E(dx, y)$ | $\tilde{Y} \xrightarrow{\kappa_{Y\tilde{Y}}} Y \xrightarrow{E} X \xrightarrow{\kappa_{X\tilde{X}}} \tilde{X}$ | $\left(\kappa_{X\tilde{X}}(\kappa_{Y\tilde{Y}}\tilde{\ell} \circ \tilde{H}), P\right) \equiv_{BR} \left(\tilde{\ell} \circ \tilde{H}, \kappa_{Y\tilde{Y}}\pi_Y \times \kappa_{X\tilde{X}}E_{\tilde{Y}}\right)$ |
| S-$\tilde{X}$, 2D-$\tilde{Y}$ | $\tilde{P} = (\kappa_{Y\tilde{Y}} * \kappa_{X\tilde{X}}) \circ (\pi_X \times F)$ $= \int_X \kappa(d\tilde{x}, x)\pi_X(dx)\sum_Y \kappa(d\tilde{y}, y)F(dy, x)$ | $\tilde{X} \xrightarrow{\kappa_{X\tilde{X}}} X \xrightarrow{F} Y \xrightarrow{\kappa_{Y\tilde{Y}}} \tilde{Y}$ | $\left(\kappa_{X\tilde{X}}(\kappa_{Y\tilde{Y}}\tilde{\ell} \circ \tilde{H}), P\right) \equiv_{BR} \left(\tilde{\ell} \circ \tilde{H}, \kappa_{X\tilde{X}}\pi_X \times \kappa_{Y\tilde{Y}}F_{\tilde{X}}\right)$ |
| S-$\tilde{X}$, 1D-$\tilde{Y}$ | $\tilde{P} = (\kappa_{X\tilde{X}} * \kappa_{XY\tilde{Y}}) \circ (\pi_X \times F)$ $= \int_X \kappa(d\tilde{x}, x)\pi_X(dx)\sum_Y \kappa(d\tilde{y}, x, y)F(dy, x)$ | $\tilde{X} \xrightarrow{\kappa_{X\tilde{X}}} X \xrightarrow{F} Y \xrightarrow{\kappa^{\tilde{Y}}} \tilde{Y}$ $\xdashrightarrow{\kappa^{\tilde{Y}}}$ | $\left(\kappa_{X\tilde{X}}(\kappa_{XY\tilde{Y}}\tilde{\ell} \circ \tilde{H}), P\right) \equiv_{BR} \left(\tilde{\ell} \circ \tilde{H}, \kappa_{X\tilde{X}}\pi_X \times \kappa_{XY\tilde{Y}}F_{\tilde{X}}\right)$ |
| 2D-$\tilde{X}$, S-$\tilde{Y}$ | $\tilde{P} = (\kappa_{Y\tilde{Y}} * \kappa_{XY\tilde{X}}) \circ (\pi_Y \times E)$ $= \sum_Y \kappa(d\tilde{y}, y)\pi_Y(dy)\int_X \kappa(d\tilde{x}, x, y)E(dx, y)$ | $\tilde{Y} \xrightarrow{\kappa_{Y\tilde{Y}}} Y \xrightarrow{E} X \xrightarrow{\kappa^{\tilde{X}}} \tilde{X}$ $\xdashrightarrow{\kappa^{\tilde{X}}}$ | $\left(\kappa_{XY\tilde{X}}(\kappa_{Y\tilde{Y}}\tilde{\ell} \circ \tilde{H}), P\right) \equiv_{BR} \left(\tilde{\ell} \circ \tilde{H}, \kappa_{Y\tilde{Y}}\pi_Y \times \kappa_{XY\tilde{X}}E_{\tilde{Y}}\right)$ |
| 1D-$\tilde{X}$, 1D-$\tilde{Y}$ | $\tilde{P} = (\kappa_{XY\tilde{X}} * \kappa_{Y\tilde{Y}}) \circ (\pi_Y \times E)$ $= \sum_Y \kappa(d\tilde{y}, y)\pi_Y(dy)\int_X \kappa(d\tilde{y}, x)E(dx, y)$ | $\tilde{X} \xrightarrow{\kappa_{X\tilde{X}}} Y \xrightarrow{E} X \xrightarrow{\kappa_{XY\tilde{Y}}} \tilde{Y}$ | $\left(\kappa_{Y\tilde{X}}(\kappa_{XY\tilde{Y}}\tilde{\ell} \circ \tilde{H}), P\right) \equiv_{BR} \left(\tilde{\ell} \circ \tilde{H}, \kappa_{Y\tilde{X}}(\pi_Y \times \kappa_{XY\tilde{Y}}E)\right)$ |
| 1D-$\tilde{X}$, 1D-$\tilde{Y}$ | $\tilde{P} = (\kappa_{XY} * \kappa_{X\tilde{X}}) \circ (\pi_X \times F)$ $= \int_X \kappa(d\tilde{y}, x)\pi_X(dx)\sum_Y \kappa(d\tilde{x}, y)F(dy, x)$ | $\tilde{Y} \xrightarrow{\kappa_{X\tilde{Y}}} X \xrightarrow{F} Y \xrightarrow{\kappa_{Y\tilde{X}}} \tilde{X}$ | $\left(\kappa_{Y\tilde{X}}(\kappa_{X\tilde{Y}}\tilde{\ell} \circ \tilde{H}), P\right) \equiv_{BR} \left(\tilde{\ell} \circ \tilde{H}, \kappa_{X\tilde{Y}}(\pi_X \times \kappa_{Y\tilde{X}}F)\right)$ |
| 1D-$\tilde{X}$, 2D-$\tilde{Y}$ | $\tilde{P} = (\kappa_{Y\tilde{X}} * \kappa_{XY\tilde{Y}}) \circ (\pi_Y \times E)$ $= \sum_Y \kappa(d\tilde{x}, y)\pi_Y(dy)\int_X \kappa(d\tilde{y}, x, y)E(dx, y)$ | $\tilde{X} \xrightarrow{\kappa_{Y\tilde{X}}} Y \xrightarrow{E} X \xrightarrow{\kappa_{XY\tilde{Y}}} \tilde{Y}$ $\xrightarrow{\kappa_{XY\tilde{Y}}}$ | $\left(\kappa_{Y\tilde{X}}(\kappa_{XY\tilde{Y}}\tilde{\ell} \circ \tilde{H}), P\right) \equiv_{BR} \left(\tilde{\ell} \circ \tilde{H}, \kappa_{Y\tilde{X}}(\pi_Y \times \kappa_{XY\tilde{Y}}E)\right)$ |
| 2D-$\tilde{X}$, 1D-$\tilde{Y}$ | $\tilde{P} = (\kappa_{XY} * \kappa_{XY\tilde{X}}) \circ (\pi_X \times F)$ $= \int_X \kappa(d\tilde{y}, x)\pi_X(dx)\sum_Y \kappa(d\tilde{x}, x, y)F(dy, x)$ | $\tilde{Y} \xrightarrow{\kappa_{X\tilde{Y}}} X \xrightarrow{F} Y \xrightarrow{\kappa_{XY\tilde{X}}} \tilde{X}$ $\xrightarrow{\kappa_{XY\tilde{X}}}$ | $\left(\kappa_{XY\tilde{X}}(\kappa_{X\tilde{Y}}\tilde{\ell} \circ \tilde{H}), P\right) \equiv_{BR} \left(\tilde{\ell} \circ \tilde{H}, \kappa_{X\tilde{Y}}(\pi_X \times \kappa_{XY\tilde{X}}F)\right)$ |
| 2D-$\tilde{X}$, 2D-$\tilde{Y}$ | No integral can be isolated, all priors and posteriors are affected by both factors. | $\tilde{Y} \xrightarrow{\kappa_{XY\tilde{Y}}} Y \xrightarrow{E} X \xrightarrow{\kappa_{XY\tilde{X}}} \tilde{X}$ $\xrightarrow{\kappa_{XY\tilde{Y}}}$ $\tilde{X} \xrightarrow{\kappa_{XY\tilde{X}}} X \xrightarrow{F} Y \xrightarrow{\kappa_{XY\tilde{Y}}} \tilde{Y}$ $\xrightarrow{\kappa_{XY\tilde{X}}}$ | No result using the factorization. |

# Supplementary References

[30] George Shackelford and Dennis Volper. Learning k-dnf with noise in the attributes. In *Proceedings of the first annual workshop on Computational learning theory*, pages 97–103, 1988.

[31] Sally A. Goldman and Robert H. Sloan. Can pac learning algorithms tolerate random attribute noise? *Algorithmica*, 14(1):70–84, 1995.

[4] Xingquan Zhu and Xindong Wu. Class noise vs. attribute noise: A quantitative study. *The Artificial Intelligence Review*, 22(3):177, 2004.

[19] Robert C Williamson and Zac Cranko. Information processing equalities and the information-risk bridge. *arXiv preprint arXiv:2207.11987*, 2022.

[5] Donald B Rubin. Inference and missing data. *Biometrika*, 63(3):581–592, 1976.

[6] Roderick JA Little and Donald B Rubin. *Statistical analysis with missing data*, volume 793. John Wiley & Sons, 2019.

[5] Nagarajan Natarajan, Inderjit S Dhillon, Pradeep K Ravikumar, and Ambuj Tewari. Learning with noisy labels. *Advances in neural information processing systems*, 26, 2013.

[34] Giorgio Patrini, Alessandro Rozza, Aditya Krishna Menon, Richard Nock, and Lizhen Qu. Making deep neural networks robust to label noise: A loss correction approach. In *Proceedings of the IEEE conference on computer vision and pattern recognition*, pages 1944–1952, 2017.

[7] Brendan Van Rooyen and Robert C Williamson. A theory of learning with corrupted labels. *J. Mach. Learn. Res.*, 18(1):8501–8550, 2017.

[32] Dana Angluin and Philip Laird. Learning from noisy examples. *Machine Learning*, 2:343–370, 1988.

[33] Avrim Blum and Tom Mitchell. Combining labeled and unlabeled data with co-training. In *Proceedings of the eleventh annual conference on Computational learning theory*, pages 92–100, 1998.

[12] Brendan Van Rooyen, Aditya Menon, and Robert C Williamson. Learning with symmetric label noise: The importance of being unhinged. *Advances in neural information processing systems*, 28, 2015.

[13] Takashi Ishida, Gang Niu, Weihua Hu, and Masashi Sugiyama. Learning from complementary labels. *Advances in neural information processing systems*, 30, 2017.

[14] Takashi Ishida, Gang Niu, Aditya Menon, and Masashi Sugiyama. Complementary-label learning for arbitrary losses and models. In *International Conference on Machine Learning*, pages 2971–2980. PMLR, 2019.

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

[41] Saunders Mac Lane. *Categories for the working mathematician*, volume 5. Springer Science & Business Media, 2013.

[42] Erhan Çinlar. *Probability and Stochastics*. Springer, 2011.

[20] Erik Torgersen. *Comparison of statistical experiments*, volume 36. Cambridge University Press, 1991.