# OpenReview forum: "A General Framework for Learning under Corruption: Label Noise, Attribute Noise, and Beyond"
_NeurIPS.cc/2023/Conference — Submitted to NeurIPS 2023_

### Official Review · Reviewer_5hDq · 2023-07-05

**Soundness:** 3 good
**Presentation:** 2 fair
**Contribution:** 3 good
**Rating:** 5
**Confidence:** 2

**Summary:**

Even though there have been a lot of studies on learning in noisy settings, they are often disparate, and the authors aim to present a unified view of a diverse class of corruptions and their implications on learning.

**Strengths:**

- The paper offers an interesting direction of looking at existing learning formulations in a noisy setting. It attempts to unify a more extensive class in a single umbrella and then analyzes the different settings through the unified framework.
- They propose a new taxonomy of corruption in classification sitting and attempt to link the existing works to the taxonomy.
- The authors also derive a loss correction formula, which would help infer the consequences of different types of corruption on learning.

**Weaknesses:**

I encourage the authors to discuss the practical implications of the paper's theory, which would strengthen the submission.
- It would also be helpful to understand how much insight their work provides compared to the corresponding works for each problem. For instance, what advantages does unifying the noises & studying them offer as opposed to studying them in separation?
- One of the issues is how realistic the assumption is that we know the process, which adds to the corruption. Does the Markov kernel capture all the classes of often encountered corruptions (or what assumptions does it impose on each class of noises)?
- In practice, the models often pick up spurious correlations in the data. Does that also fit into the taxonomy considered? How does the bias problem (regarding the fairness of predictors) fit into the framework (even if that can be considered a noise-generating process)?

**Questions:**

Refer to weaknesses

**Limitations:**

The authors do talk about limitations, but it isn't complete. I encourage the authors to refer to the questions & address them in the list of limitations.

---

> ### Author Rebuttal · Authors · 2023-08-09
>
> **Insights of the unification framework.** See the global response.
>
> **Assumption of the known corruption process.** Indeed, one typically does not (and can not) know the specific corruption at hand. However, for our purposes (making a taxonomy and comparing different types of corruption) we of course have to assume that they are known.  Our hope is that by formulating the types of corruption more abstractly, we may be able to develop better tools to not only deal with corruptions when they occur, but also to help identify which ones are present in particular situation.
> The situation is in fact even worse than the reviewer notes. Even if the corruption model is specified by an oracle, the corruption parameters are not estimable in general. They can be estimated in some special cases with case-dependent assumptions, for example, in the case of CCN, “perfect examples” of each class are assumed for the estimation (see Theorem 3 in [34]); in the case of MCD, the true class-conditional distributions are assumed to be mutually irreducible for the estimation (see Proposition 8 in [50]). Analysis of such conditions for every corruption scenario is beyond the scope of the current paper and is therefore reserved for future investigations.
>
> **Spurious correlation as corruption.** This is a very fine point, and one which we have also pondered. We do know that there are some “corruptions” which cannot be modeled by stochastic kernels; but we do not have a comprehensive story as to which ones. Whether things like “do-interventions” or fairness violations or even spurious correlation can be understood as forms of corruptions is a very intriguing question and one which we will pursue in the future.

---

> > ### Comment · Reviewer_5hDq · 2023-08-22
> >
> > Thank you for the detailed response, after looking at the rebuttal and the other reviews, I will keep my scores.

---

### Official Review · Reviewer_FFYg · 2023-07-07

**Soundness:** 3 good
**Presentation:** 1 poor
**Contribution:** 2 fair
**Rating:** 4
**Confidence:** 2

**Summary:**

The paper presents a theoretical framework for analyzing corruptions in machine learning using Markov kernels. Within this framework, the authors establish a taxonomy of various types of corruptions and provide data processing equalities for each category. Additionally, the paper derives corrected loss functions for different types of corruptions within their theoretical framework.

**Strengths:**

The authors categorize corruptions in a rigorous and systematic manner. This classification and the theoretical view from Markov kernels can potentially help future research future research in understanding and studying different types of corruptions. The derived corrected loss can potentially be used to improve performance of machine learning algorithms in the presence of corruptions.

**Weaknesses:**

Overall, I find it hard to appreciate the contribution of the paper. It lacks clear intuition and practical implications, making it difficult to understand its usefulness. Theorems are presented without concrete insights, and it is unclear how the results can guide real-world practice.
1. In Section 4, the paper discusses the "consequences" of corruption in supervised learning. However, there is a lack of concrete interpretations of these consequences. How do Theorems 3, 4, 5 translate to real-world machine learning scenarios? What can we deduce about the nature of these corruptions and their implications?
2. The paper introduces a corrected loss function, but it is not well-explained how it operates or why it is effective. What is the underlying intuition behind this correction and how does it improve the performance under label noise?
3. It would be beneficial if the authors could provide concrete examples to illustrate how the corrected loss function can be implemented. For instance, what would be the formula of the loss in the context of cross-entropy loss under label noise? Right now it is even unclear whether these losses are feasible and whether they require any information that is not observable.
4. It would also be valuable to include experiments that demonstrate the effectiveness of the derived corrected loss function.

**Questions:**

See questions raised in Weaknesses.

**Limitations:**

The authors have adequately discussed some limitations in section 6. For other limitations, see Weaknesses.

---

> ### Author Rebuttal · Authors · 2023-08-09
>
> **Contributions of the paper.** See the global response.
>
> **On the Data Processing Equality theorems.** The unification framework allows us to compare the consequences of different corruptions on learning by analyzing their resulting changes in the Bayes risk. Specifically, we establish equivalence between the clean and corrupted learning problems through the lens of the Bayes risk metric and demonstrate Data Processing Equality (DPE) results in Theorem 3, 4, 5, and Corollary 6.
>
> These findings reveal a crucial conceptual distinction: when only the labels are corrupted, such as class-conditional label noise, instance-dependent label noise, etc., only the loss function is affected, while the model class remains unaffected by the corruption kernel. However, in more complex cases involving corruption also on covariates, such as combined simple noise, concept drift, etc., both the loss function and the model class are influenced by the corresponding factorization of the corruption kernel.
>
> As a result of the data processing equalities, the effective model class is reduced or enlarged in these cases, which had been overlooked until the first DPE results provided inspiration for this insight [19], and is a prominent result for understanding the consequences in real-world scenarios.
>
> Furthermore, the DPE results indicate that correcting complex corruptions is significantly more complicated than dealing with corruptions that solely affect labels, because the cleaning kernel must act on both the loss and the hypothesis; the correction is also dependent on additional parameters when corruptions are not simple (refer to the next point for details).
>
> **The Data Processing Inequality and our loss correction.** Our loss corrections rely on the DPE theorems proved for different types of corruptions, ensuring that the performance on a well-defined corrupted learning problem will be the same, in terms of Bayes risk, of the clean one. The Data processing inequality (DPI) states that $BR_{\ell \circ \mathcal{H}} ( P ) \le BR_{\ell \circ \mathcal{H}} (\tilde{P})$, which means that the performance of the corrupted problem is always worse or equal than the clean one, according to the same minimization set $\ell \circ \mathcal{H}$.
>
> The equality ensures that whatever correction we find that respects the DPE will give us a better performance than the initial minimization problem, specifically the same as we were learning on the clean one. Therefore, its *effectiveness* is proved to be true by Theorem 8 on the optimum of the learning problems.
>
> As for examples of how the loss function will be implemented, we will add an example explaining the difference between simple and dependent noise, and how this changes the loss correction computation. The interpretation of the loss function in the $Y$ corruption cases is simple, because it corresponds to a simple loss reweighting for each label value. In the $X$ corruption case, and therefore in the joint cases, the intuition is harder to get because it involves computing the expectation over the set of *reachable predictions from the considered* $\tilde{h}^*$ through the cleaning kernel $\kappa^{+}$.
>
> To better grasp this, we can consider the following finite case example that will be added to the manuscript. Assume both the $X, Y$ spaces to be finite with respective size $m, n$. The push forward measure of point 2 of Theorem 8, i.e. $\kappa^{ X}h(\tilde{x})$, can be graphically represented as in Figure S1 in the attached pdf. The figure can be interpreted as in the following: we fix a point $\tilde{x}$ in $X$, the input of our stochastic process; we then obtain different probabilities $\kappa^X_i \coloneqq \kappa^X(\tilde{x}, \{ x_i \})$ of “jumping” (in the sense of Markov chains) from $\tilde{x}$ to another point $x_i \in X$; lastly, for each reachable point $x_i$ we have another stochastic process, induced by $h$, deciding an output point of kind ${y_j}^{(x_i)}$  $\sim h(x_i, \{ y_j\} )$. The process described differs from chaining Markov kernels, because here we are not averaging on the possible outcomes of the first stochastic process, i.e. which $x_i$ we might jump to.
>
> The visual representation of a reachable prediction from $\tilde{h}^*$ as given in Figure S1 is, in words, a two-fold stochastic process. It can be thought of as the distribution associated with $\kappa' \coloneqq$ ( $\kappa^X \tilde{h}^*$ ) evaluated on the point $ (\tilde x,  \mathsf{X}) \in X \times X$, a random variable itself equal to a Markov kernel parameterized by a random variable.
> This induces a set of random variables, the set of $\kappa^X$-reachable predictions from $\tilde{h}^*$ when we consider the optimal hypothesis, equal to { $\kappa_{\tilde x, \mathsf X}' \ \text{s.t.} \ \tilde{x} \in X ,  \mathsf{X} \ \text{r.v. on} \ X $ } $\subseteq \mathcal{P}(Y)$.
> The correction via reachable predictions is clearly harder to implement and we did not propose a method for it, given the scope of the current paper. We aimed to underline how the correction gets more complex even in a S-$\tilde{X}$ case, and stimulate further research in this direction.

---

> > ### Comment · Reviewer_FFYg · 2023-08-21
> > **Thanks for your response**
> >
> > Thank you for your detailed explanation, and I apologize for the delayed response. I now recognize the contributions numbered 1, 2, and 4 in the global rebuttal. However, I remain uncertain about how this work can effectively inform the development of principled methods for learning under corruption (contribution 3). For example there still lack concrete examples of how to actually use the corrected loss and the authors did mention the difficulty of using it in their responses to other reviewers and in the rebuttal. For the results on DPE, providing more detailed insights into how the conclusions about the enlarged or reduced effective model class could be practically applied would be beneficial. Meanwhile I personally think that the topic of learning under corruption is particularly geared towards practical applications, aiming to guide the design of new learning algorithms, unlike other contexts where the goal is to build theoretical understanding of certain phenomenon observed in practice. Papers in this category could benefit from emphasizing clear takeaways for practitioners, which are somewhat lacking in the current paper. That being said, I am not very familiar with the theoretical tools used here and would like to leave it for the AC and other reviewers to evaluate the theoretical contribution. It's possible that the theoretical value might outweigh the lack of clear practical insights. Due to this consideration, I've raised my score slightly and reduced my confidence level.

---

### Official Review · Reviewer_2zzP · 2023-07-07

**Soundness:** 3 good
**Presentation:** 2 fair
**Contribution:** 2 fair
**Rating:** 3
**Confidence:** 3

**Summary:**

This study addresses the prevalent issue of data corruption in machine learning and the lack of a unified understanding of how different corruption models interrelate. The authors analyze corruption models at the distribution level using a comprehensive framework based on Markov kernels, providing a fresh perspective on the issue. They draw attention to joint and dependent corruptions on both labels and attributes, which existing research often overlooks. By examining the effects of these corruptions on supervised learning through Bayes Risk changes, the authors gain insights into the implications of complex corruptions. The proposed framework also has applications in corruption-corrected learning, which the authors explore in this paper. However, the paper does not have any empirical contributions, and has several theoretical discussions that need further clarifications.

**Strengths:**

1. The paper delves into the crucial problem of data corruption in machine learning, offering a comprehensive framework based on Markov kernels to analyze corruption models at the distribution level.

2. It paves a way for future quantitative comparisons by delivering a foundational understanding of more complex corruptions' consequences.

3. The authors' taxonomy of corruption is comprehensive and organized hierarchically through the notion of dependence, helping to relate existing corruption models.



**Weaknesses:**

1. the paper does not involve any empirical studies, which is very rare for this venue, although there are precedents. Most theoretical papers still choose to validate their finding with some small-scale experiments, the authors does not seem to clarify a particular reasons that they do not need any empirical evidence to support their findings.

    - This seems a particular issue when the authors explicitly suggest that the theoretical framework can help answer "what can we do to ensure unbiased learning from biased data" (line 228), and mentions two previous works [7] and [34], where [34] has a decent amount of empirical works.

2. on the theoretical end, there are several questions need further clarifications, potentially paint a significant issue of rigor of the results. (see below questions)

**Questions:**

1. there are several parts of the theoretical discussions need further clarifications that potentially paint a challenge to the rigor of the discussed results.

    - In the manuscript, there are two assumptions discussed A1 at line 259, and A2 at line 265. However, such assumptions are used again in appendix (line 746, A1-A3)

         - (minor) it seems the appendix are re-using the notations A1, A2 to discuss different assumptions, and certain naming conventions need to be avoided to improve the rigor of the work
         - (major) the appendix proof suggests that the proved results Theorems are proved under assumptions A1-A3, yet the main theorem body in the main manuscript has no such mentions of the assumptions (within theorem body), which could potentially mislead future readers.
         - Also, A1 in appendix does not seem to be a conventionally rigorious way of stating an assumption.

2. As the theorem body are usually preferred to be self-contained, it seems not so conventional to use footnote to explain notations.


3. line 262-265 seems very confusing, the first part seems to suggest we need $h'\neq h^\star$, and the second half assumes $h' =  h^\star$ as A2.
     - if I understand correctly, A2 is a very strong assumption, it might be crucial to talk about its practical implications. (also, another reason that some empirical evidence will be needed.)

4. if possible, please try to talk about the practical implications of all the assumptions.

**Limitations:**

no explicit discussions of limitations, but there are discussions about future directions, which might be interpreted as the limitation discussion.

---

> ### Author Rebuttal · Authors · 2023-08-09
>
> **Lack of empirical studies on the corrected losses.** Please refer to the global response for answers to the general points. As explained, the main contribution of this paper is an exhaustive framework encompassing all possible pairwise corruptions, along with theoretical analyses of their consequences and loss corrections. Observe that [34] solely focuses on studying loss correction for a simple corruption instance in our framework, i.e., class-conditional noise (CCN) as a S-$\tilde Y$ corruption. In contrast, we proved loss correction results for all corruption instances within the framework.
>
> As a result, empirically testing the corrected loss in [34] is relatively straightforward, while for us, it poses certain technical difficulties. The corrected loss for the simple $Y$ corruption in [34] is a subcase of the first point in our Theorem 8, where the cleaning kernel only acts on the loss. However, in more complex cases where $X$ is also corrupted (points 2, 3, 4 in Theorem 8), the cleaning kernel acts on both the loss and the hypothesis, leading to a corrected loss that involves approximating expectations over the set of all the reachable hypotheses and therefore hard to implement.
>
> **Discussions on assumptions A1-5.** We accept that relegating the additional two technical assumptions in the appendix is a source of confusion, and we have made them all visible in the main body now.
>
> We reformulate A1 as “We assume the loss function to be positive, proper and bounded” and restate the last two here as a reminder for the reader:
>
> A4. We assume the existence of an invertible hypothesis $\tilde{h}^* \in \mathcal{H}$ (in the sense of function);
>
> A5. We ask $\tilde{h}^*$ to satisfy the equation $\kappa^{\dagger}(\ell \circ \tilde h^* ) = \tilde{\ell} \circ \tilde h^*$.
>
> Recall that $\kappa^{\dagger}$ is the pseudo-inverse of the kernel and that the   $\tilde{}$  symbol indicates the part of the corrupted learning problem. The hypothesis $\tilde{h}^*$ is the optimal one in the corrupted space, in general different from the $h^*$ optimal on the clean problem. For the loss correction theorem, we did not assume $h' = h^*$ as mentioned in the review (Question 3). As stated in line 265, we are asking for $h' = \tilde{h}^*$, while underlying in line 263 that assuming $h' = h^*$ would make the corruption trivial.
>
> The assumption A4 is standard. Different works in the literature ask to learn invertible models. Nevertheless, there might be weaker conditions enabling loss correction, but we did not explore them here (line 260). As for assumption A5, it is not such a strong assumption: choosing a particular $\tilde{h}^*$ has the sole effect of inducing a different $\tilde{\ell}$, given that the $\circ$-factorization of the corrupted minimization space is not unique. Therefore, we are only fixing a corrupted learning problem that would otherwise be not unique.
>
> Assumption A5 is practically focusing on the effect of the corruption at the optimum solely on the loss function, instead of involving the corrupted optimum $\tilde{h}^*$. To clarify what that means, consider a stronger assumption of limiting the effect of corruption on the loss for all hypotheses in the model class. In this case, the existence of a single loss working for all hypotheses is unclear, so we restrict the assumption to the optimum only.

---

### Official Review · Reviewer_uSg2 · 2023-07-07

**Soundness:** 3 good
**Presentation:** 2 fair
**Contribution:** 3 good
**Rating:** 6
**Confidence:** 3

**Summary:**

The paper presents a framework for systematically analyzing and categorizing corruption models in machine learning. The authors propose a new taxonomy of corruption based on its dependence on the feature and label space, rather than relying on invariance assumptions. They utilize the concept of Markov kernels to provide an exhaustive framework that includes all possible pairwise stochastic corruptions. The authors also derive corruption-corrected loss functions within the framework.

**Strengths:**

- A novel unification of existing data corruption models: The proposed theoretical model based on the Markov kernel is capable of modeling various existing data corruption and distribution shifts. This offers a new perspective to understand how different data dynamics connect to each other.

- Insights on how to correct the errors: The authors also derive loss correction formulas for a few instances of the unified data corruption model. These results give an abstract way to study data corruption-aware learning.

**Weaknesses:**

- Unaccessible to a wider community: One major concern with this paper is whether it is accessible to the NeurIPS community. This paper seems to be a pure theory paper: it is symbol-heavy, theory-oriented, and without any empirical support. The theory community (e.g., COLT) might appreciate it more, and I am a bit worried about the NeurIPS community. See my questions in the next section for more details.

- Unclear concrete applications of the proposed theoretical models: It is unclear how useful the unification is. Do its insights give statistically or computationally more efficient algorithms for corruption-aware learning? Or does it implies one corruption instance subsume another and thus focusing on it is enough? Or does it reveal new corruption instances not studied before but important in practice? A good paper, in my opinion, should give useful advice to its readers.

**Questions:**

A few concrete questions regarding the weakness points.

- How does the Kernel parameter correspond to the original parameter in the existing corruption model? For example, the authors claim that 1D-X, 2D-Y correspond to target shift/label shift. Given the label shift matrix p(y=j|y=i), how does the markov kernel look like?

- How is the derived corruption-aware learning loss different from the existing method? For example, [49] proposes a correction algorithm for label shift with sample complexity guarantee. How to compare [49]'s algorithm with the derived approach? When should which be used?

- Could you give a running example throughout the paper?  For example, give a concrete label shift example (with the shift matrix), show its corresponding kernel, compare the corresponding corrected loss with existing correction algorithms, and justify why the new techniques are desired.

- How does the proposed model connect to modern data distribution shifts? For example, sparse joint shift [r1] was a recently proposed distribution shift model measuring the joint shift of covariate and label. What is the Markov kernel for this model? Where should it be in Table 1 and Figure 1? Does the kernel model reveal new connections between sparse joint shifts and more traditional shifts such as covariate shifts?

[r1] Chen et al, Estimating and Explaining Model Performance When Both Covariates and Labels Shift, 2022.

- More generally, what are some new data corruption instances offered by the new model? Could you clearly state how it is different from the existing ones, and why it might be encountered in practice?

**Limitations:**

Yes

---

> ### Author Rebuttal · Authors · 2023-08-09
>
> **Accessibility and applications of the work.** See the global response.
>
> **Label shift as Markov kernel.** We believe the reviewer’s concern arises simply from the non-standardized naming of corruptions in the literature.
> The label shift matrix described in the question is called class-conditional noise (CCN) in the literature [5, 6, 34], which is for us a S-$\tilde{Y}$ corruption. As explained in the main, Markov kernels are a generalization of conditional probabilities; for $Y$ discrete, it is equal to the conditional probability, so $\kappa_{Y \tilde{Y}} \coloneqq [ P(\tilde{\mathsf Y}=i | \mathsf Y=j) ]_{ij}$.
>
> The label shift proposed in [49] additionally requires keeping the experiment $E( dx | i) = P(\mathsf X = x | \mathsf Y = i)$ to be invariant and is known as *target shift.* Notice that the equality between $E$ and the conditional distribution holds in the case of  $X$ discrete.
>
> **Comparison with the loss correction algorithm in [49].**  [49] assumes a corruption of the distribution $P$ into $Q$, requiring $Q \ll P$ (absolute continuity). That implies having a selection bias kind of corruption, which we show can be represented either as a non-normalized diagonal positive matrix $[\delta_{ij} P(\mathsf{Y_Q}=i| \mathsf{Y_P} = j)]_{ij}$ or as a Markov kernel (stochastic matrix) $[P((\mathsf{Y_Q}=i| \mathsf{Y_P} = j)] _{ij}$ (see Section S2.3 in the Supplementary for definition and analysis of selection bias).
>
>
> [49] chooses the non-normalized diagonal matrix representation as it is the more natural, and computes the importance reweighting estimator $\tilde{\ell} = w \ell$ for loss correction. To avoid accessing the distribution $Q$, they compute $w = [P(f(\mathsf X_P)=i | \mathsf Y_P=i)P(\mathsf Y_P=i)]_{ij}^{-1} [P(\mathsf Y_Q=i)]_i$ , with $f$ being a labeling function. Their method is then further expanded into an algorithm to estimate such $w$ from data, only having access to the observed frequencies of the predictor $f$ (possibly sub-optimal) against the true labels.
>
> In our work, we choose the Markov kernel representation. Our computations are similar in the sense of correcting the loss using some knowledge about the corruption process. However, it is unclear if in general the correction induced by the $w$ representation is the same induced by the Markov kernel. Nevertheless, the expected risk of the two corrections will be the same at the optimum.
>
> **New data corruption instances and connection with [r1].** Note that the proposed framework is exhaustive in modeling pairwise corruptions between two probability spaces. Within this framework, some corruption instances correspond to well-studied cases with established names, e.g., covariate shift as at least a S-$\tilde{X}$ and at most a (2D-$\tilde{X}$, 2D-$\tilde{Y}$) corruption, CCN as a S-$\tilde Y$ corruption. However, many of the other corruptions remain less explored, for example, the combined simple noise (S-$\tilde{X}$, S-$\tilde{Y}$) corrupting both covariates and labels that is an extension of the famous CCN is rarely investigated, but its consequences and corrections are more complicated than a S-$\tilde Y$.
>
> The above-mentioned sparse joint shift (SJS) [r1] is also an example in this regard. It can be modeled at least as a S-$\tilde{X}_{I}$ or a S-$\tilde{Y}$, and at most as a (2D-$\tilde{X}$, 2D-$\tilde{Y}$) corruption, in both cases such that the partial experiment $p(x _{I_c}| x _{I}, y)=E(x _{I_c}| x _{I}, y)$ is left invariant. This is similar to how we consider the connection to traditional taxonomies, for example, what leads to drawing the connection with covariate shift and more — see Table S1 in the attached pdf. This way of viewing the SJS corruption clearly places it as a case in-between CCN and a subcase of the simplest covariate shift, i.e., the S-$\tilde{X}\coloneqq$ (S-$\tilde{X} _{I}$, $\delta _{X _{I _c}}$).
>
> We further underline that although these corruptions have distinct established names, our proposed standardized naming system provides a clear understanding of their differences by explicitly noting their dependent parameters and the corrupted variables, which are left undefined in the traditional notations.
>
> **On the practical choice of the corruption instances.** We cannot be certain about what is encountered in practice as we do not investigate testing a corruption model; there are no guarantees regarding its existence or absence in the data generation (this weakness is shared by almost all the current literature). We do provide a comprehensive list of possible more complex corruption models that theoretical researchers and practical practitioners have a broader playing ground to explore, which would not be considered if not formally stated.

---

### Author Rebuttal · Authors · 2023-08-09

Thanks to the reviewers for their response. In this global response, we deal with some common themes across their reviews. We address more specific questions in separate responses.

We think a number of the concerns of the reviewers are a result of not understanding our intent. Indeed, we have no experimental results. Indeed, we do not offer any turn-key solutions.

But that was not our goal. We observed the continued proliferation of ever more complex models of corruption, along with a panoply of techniques designed to mitigate the corruptions. Rather than further contributing to this zoo, we have attempted an initial taxonomy of it.

No, we have not covered everything yet. But yes, we have covered much, and indeed that gap between what we can incorporate, and what is still not understood is quite large.

No, there is no single running example, but that is because we are discussing, in a unified manner, a very large number of types of corruption: a single running example is antithetical to our whole enterprise.

Nevertheless, we see significant value in the exercise we have attempted, because it allows us to see commonalities between the different models of corruptions, it allows us to transcend the confusing bespoke terminology different authors use, and crucially it allows us to sensibly grapple with the problem that in practice you do not (and often cannot) know exactly what corruption your data has undergone.

**Regarding some specific concerns:**

Indeed many papers at NeurIPS have experiments, but they make no sense for the purpose we have set ourselves (that of building a well-grounded taxonomy of corruption).

No, we do not have pragmatic turn-key solutions; that is because the state of understanding needed to provide these (in general is simply absent). Thus rather than proposing yet another potential mitigation, whose practical success essentially depends on correctly modeling the particular corruption at hand, we have focussed our energies on making an initial map of the territory, before making any attempt to conquer it.

Yes, the paper uses terminology and notation unfamiliar to the NeurIPS community, but we note that concern would have held true for every single idea in machine learning at some point. We think it a feature of our paper that is different from the other 10,000 submissions, not a bug.

And finally no, it will not be of interest to everyone at NeurIPS, but we believe that there is no such paper ever in the history of NeurIPS and thus it is unreasonable to demand it of us.

**Some more specific points:**

Almost all existing work on data corruption studies specific models of corruption and offers partial mitigations. Such an approach cannot, even in principle, answer questions involving the comparison of different types of corruption. Our framework does offer this possibility.

In the absence of a general framework, there have been blind spots in the literature regarding some corruption models that can well arise empirically; our systematic framework allows us to capture a wider range of corruptions.

A final general point is that the reviewers appeared to be of the view that any contribution to the literature must be a complete story, in the sense that the idea of the paper, in itself, can solve a problem. It is true that many NeurIPS papers have this pattern. We see this as more of a weakness than a strength because it necessarily means that the authors of such papers limit their ambition to that which they can entirely complete in one paper.

Instead, we believe that science accumulates over a long time scale, and we are offering a building block, not a building. But it is a building block that is currently absent, and one which seems essential for further progress on the question of data corruption.

**Summary of contributions:**

- *First exhaustive framework of pairwise corruptions.* In data corruption research, papers often assume a specific corruption model and suggest a tailored algorithm, posing two key challenges to the advancement of the field — namely, whether the proposed models characterize all types of corruption and how they relate. As a first step to explore these fundamental questions, we propose a comprehensive corruption framework.

- *Offer qualitative comparisons of the consequences of various corruptions.* We compare how different corruption instances in the framework affect learning by analyzing their resulting changes in Bayes Risk. Theorem 3, 4, 5, and Corollary 6 reveal a crucial distinction: in the case of corruptions on $Y$, only the loss function is affected while the model class remains untouched by the corruption kernel; however, in more intricate cases involving corruptions also on $X$, both the loss function and the model class are influenced by the corresponding factorized corruption kernel.

- *Lay the groundwork for developing principled methods for corruption-corrected learning.* Thanks to the Bayes Risk results, we find a hierarchy-induced set of results on how the optimization problem changes under various corruptions and how to abstractly compute their loss corrections in Theorem 8. We ring the bell that more complex corruptions are more detrimental and require more sophisticated designs for corruption-corrected learning.

- *Provide a standardized approach for naming corruption cases.* We link known corruption models to our framework to establish a uniform nomenclature and demonstrate the relations between our model and other definitions, as shown in Table 1 and Section S1 in the Supplementary.

---

### Decision · Program_Chairs · 2023-09-21

**Decision:**

Reject

**Comment:**

The paper presents a Markov kernel based framework for analyzing and categorizing corruption models in machine learning. The paper propose a new taxonomy of corruption based on the joint distribution of labels and attributes, rather than relying on invariance assumptions. The framework uncovers existence of intricate joint and dependent corruptions on both labels and attributes and consequences of complex corruptions.

The reviewers appreciated the theoretical contribution, however the reviewers were unconvinced that the findings could have practical implications or can potentially lead to such. This was the key reason for rejection.